# Discover the Desirable Landscape Structure of Urban Parks for Mitigating Urban Heat: A High Spatial Resolution Study Using a Forest City, Luoyang, China as a Lens

**DOI:** 10.3390/ijerph20043155

**Published:** 2023-02-10

**Authors:** Kaihua Zhang, Guoliang Yun, Peihao Song, Kun Wang, Ang Li, Chenyu Du, Xiaoli Jia, Yuan Feng, Meng Wu, Kexin Qu, Xiaoxue Zhu, Shidong Ge

**Affiliations:** 1Department of Landscape Architecture, College of Landscape Architecture and Art, Henan Agricultural University, Zhengzhou 450002, China; 2College of Urban and Environmental Sciences, and Key Laboratory for Earth Surface Processes of the Ministry of Education, Peking University, Beijing 100871, China; 3International Union Laboratory of Landscape Architecture, Henan Agricultural University, Zhengzhou 450002, China; 4College of Biological Resource and Food Engineering, Center for Yunnan Plateau Biological Resources Protection and Utilization, Qujing Normal University, Qujing 655011, China

**Keywords:** urban park green space, land surface temperature, landscape structure, seasonal variations, stepwise regression, luoyang

## Abstract

Urban parks can mitigate the urban heat island (UHI) and effectively improve the urban microclimate. In addition, quantifying the park land surface temperature (LST) and its relationship with park characteristics is crucial for guiding park design in practical urban planning. The study’s primary purpose is to investigate the relationship between LST and landscape features in different park categories based on high-resolution data. In this study, we identified the land cover types of 123 parks in Luoyang using WorldView-2 data and selected 26 landscape pattern indicators to quantify the park landscape characteristics. The result shows that the parks can alleviate UHI in most seasons, but some can increase it in winter. While the percentage of bare land, PD, and PAFRAC have a positive impact on LST, AREA_MN has a significant negative impact. However, to deal with the current urban warming trend, a compact, clustered landscape configuration is required. This study provides an understanding of the major factors affecting the mitigation of thermal effects in urban parks (UP) and establishes a practical and feasible urban park renewal method under the idea of climate adaptive design, which provides valuable inspiration for urban park planning and design.

## 1. Introduction

The pace of urbanization worldwide is unprecedented, since reform and opening-up, China’s urbanization rate has increased by 45.9%, much higher than the global average in the same period [1,2]. In addition, rapid urbanization has increased heat storage in cities as more vegetated areas are replaced with underlying impervious surfaces [3], which leads to an exchange of energy between the urban surface and the atmosphere [4,5]. This has made urban land surface temperature (LST) higher than surrounding non-urbanized areas; this phenomenon is defined as an “urban heat island” (UHI) [6]. The UHI has been observed worldwide [7,8]. Industrial activities and anthropogenic heat sources have increased the scale and complexity of the UHI [9]. The intensification of UHI leads to abnormal changes in material and energy flows within cities and in the structure and function of ecosystems, so it seriously affects urban health and increases the risk of human violence and death [10,11,12]. The data from the China Meteorological Administration showed that the combined intensity of regional heat events in 2022 is the strongest since 1961, putting environmental pressure on Chinese cities [3]. Numerous empirical studies show that urban parks (UP) can cool and humidify the surrounding environment through evapotranspiration, shading, and water, playing a widely recognized leading role in reducing urban LST [13,14,15]. In the context of the background above, gaining a better understanding of the influencing factors of urban parks LST, enhancing the cooling effect, and maximizing the ecological benefits of UP is an important research direction and hot issue in improving the urban ecological environment and the city’s ability to cope with climate change and is essential to promote sustainable urban development [16,17,18].

The LST is an essential parameter for studying the urban thermal environment and is also the main index to regulate the temperature of the urban bottom air and determine the surface radiation [6,19]. What is evident from the literature is that the widely used approaches for measuring urban climate and variations in LST can be grouped into two major categories: meteorological station monitoring data and remote sensing data. 

However, due to the limited geographic coverage of meteorological stations and the complexity of dynamic heat transfer processes, it frequently provides evidence of the impact of urbanization on climate warming at regional scales [20,21]. In contrast, with the rapid development of remote sensing and geospatial science, LST based on infrared remote sensing has been widely used in quantitative urban LST due to its ease of retrieval and collection, low cost, and reusability, making it suitable for long-term continuity and related studies at different scales and accuracies [22].

The important influence of landscape structure on UHI has been widely demonstrated. An accurate assessment of land cover types is also essential for exploring the spatial pattern of UP and improving its quality [23]. Along with the advancement of remote sensing and geographic information systems, land use/land cover (LULC) data have become a major source for monitoring spatial patterns in UP, especially in studies at the city and regional levels [24]. To date, various raster-formatted LULC products have been widely applied in related research, and most of the quantitative land cover data used in the study have an accuracy of 30 m and above. Therefore, many small UP may be ignored or misidentified as urban impervious pixels and excluded from studies under the constraint of satellite pixel accuracy [25]. The landscape pattern of these parks is also hard to accurately monitor due to the limitations of resolution. In addition, small parks should not be neglected when quantifying the spatial extent and internal spatial pattern of parks, as they have been shown to have an effective cooling effect on the surrounding areas [26,27]. The distribution of landscape patterns within the parks needs to be estimated more accurately. In this paper, Word View-2 data with a resolution of 0.5 m was used to monitor the LULC data of UP and cooperate with manual visual modification to improve the accuracy of the landscape pattern of UP.

Numerous studies have used remote sensing data to explore the change mechanism and scale of LST. In terms of urban climate, urban greening is often done to offset the negative impact of impervious surfaces on urban temperatures [28]. It has been found that UP can significantly reduce urban surface temperatures [29,30]. The study of cities in different climate zones shows that the UP LST varies significantly across seasons and day and night [21,31,32]. As for driving factors, more attention has been paid to LULC, landscape structure, and energy consumption [33,34]. Many studies have confirmed that the ability of parks to regulate temperature is related to morphological indicators [35,36]. Gao found that the increased area and perimeter of parks can bring an enhanced cooling effect, and the shape index showed a relatively weak and inconsistent correlation with LST [37]. The effects of the park‘s spatial characteristics, such as the distribution, morphology, and composition of various landscapes, on the LST have also been confirmed in recent articles [38,39]. Maimaitiyiming et al. discovered that increasing the PD and ED helps reduce LST [40]. Furthermore, the composition, quantity, type, horizontal and vertical structure, and proportion of the park’s vegetation community all significantly impact the LST; trees have a stronger effect on the LST than shrubs, with herbs playing the smallest role [41,42].

Nevertheless, due to the complex interaction of various factors and the spatial heterogeneity of cities, the causes of UHI are complex and site-specific, and the relationship between landscape structure and LST is inconsistent. A single measure cannot be effectively applied to all regions [12,43] and alleviating UHI by increasing the number of parks without limitation is unattainable due to the finite urban area [44]. Liu even thinks that inferences about factors influencing LST in global cities based on a limited number of studies may be invalid [45]. On this premise, accurately quantifying the relationship between landscape structure, LST, and its specific impact trend on the regional UP and seeking more sustainable, affordable measures is a meaningful and practical way to alleviate the risk of the urban thermal environment [46]. It is also critical for targeted improvements in urban landscape management and planning, as well as understanding the urban biophysical characteristics and processes required for long-term socioeconomic and environmental decision-making [45]. Studies on UHI in China are geographically biased, mainly focusing on metropolises with high urbanization rates and economically developed coastal cities, such as Shanghai, Beijing, Shenzhen, etc. [47,48,49,50], with less focus on the less developed areas in the Central Plains and the northwest coast of China. In recent years, as one of the most crucial garden cities, forest cities, and economic and cultural centers in the Central Plains of China, Luoyang has experienced intense urbanization and rapid economic development. The city’s borders have continued to expand, and internal green space construction has continued to strengthen. At the same time, as the oldest city in the history of civilization and the eastern starting point of the Silk Road, Luoyang has a profound background in garden construction and occupies an undoubted cultural position, making it one of the most representative cities in the Central Plains. Therefore, it is imperative to explore the relationship between the landscape pattern and the LST of UP in Luoyang and find the optimal planning mode of UP for the economic and cultural construction and ecological improvement of Luoyang.

Therefore, this study conducted an accurate definition of the landscape pattern based on the high-resolution remote sensing data and a comprehensive comparison of the seasonal variation of the effects of the landscape pattern on park LST by specific park categories to put forward a concrete and implementable urban planning scheme for urban management with the focus on (1) Use high-resolution remote sensing data to obtain UP landscape types and their shape indicators accurately. (2) Retrieve LST from remote sensing satellite infrared data and probe the relationship between LST and park morphological indicators and landscape patterns. (3) Analyze the maximization mode of LST regulation in Luoyang UP and provide targeted recommendations and models for urban planning on mitigating the UHI effects through the rational allocation of the interior landscape of the park.

## 2. Materials and Methods

### 2.1. Study Area

Luoyang, the sub-city center of the Central Plains city group in China (33°21′ N–35°3′ N and 111°48′–112°35′ E) (Figure 1), is located in the Cenozoic depression basin of the loess hills in western Henan, in the middle and lower reaches of the Yellow River Basin, across the north and south banks of the Yellow River, and its total area is 15,200 km^2^ [51]. The urban history of Luoyang is over 4000 years old, and it is the birthplace of Chinese civilization. Thirteen dynasties have built their capitals here. Many gardens were built here, and it has been designated as the national landscape garden city in China (NLGCC) and the national forest city in China (NFCC). It is in the leading position in the Central Plains region regarding urban construction, optimizing urban spatial structure, and improving the ecological environment. By early 2022, the population of residents in Luoyang will be 7.069 million, and the urbanization rate will have reached 65.88%.

### 2.2. Methods

The flowchart of the study is as follows (Figure 2). First, the LST of the parks was retrieved from Landsat 8 OLI images, and parks were divided into four major categories and 17 sub-categories. Second, the LULC of the study area was classified based on the WorldView-2 images, and the landscape indicators were calculated. Based on the above data, analyze the spatial pattern and the LST of UP in different seasons, and then explore the impact trend of landscape pattern on the LST through correlation analysis and stepwise regression analysis.

#### 2.2.1. Inversion of LST

The most commonly used surface temperature inversion methods include split-window algorithms, single-channel methods, the mono-window algorithm, and the radiative transfer equation-based method (RTE) [52,53]. The RTE, also known as the atmospheric correction method, is based on subtracting the atmospheric influence on surface radiation from the total amount of thermal radiation observed by the satellite sensors to obtain the surface thermal radiation intensity, which is then converted to the corresponding LST. According to José A. Sobrino and Sekertekin [54], by RTE, the retrieved LST based on atmospheric profile measurement can reach an accuracy of 0.6 °C [55], and the results are not affected by season. Therefore, this method is introduced to retrieve the LST of Luoyang.

Seasonal LST data were derived from the thermal infrared (TIR) band for four Landsat-8 images (Row 36/Path 125) obtained from the United States Geological Survey (USGS) (https://www.usgs.gov/ (accessed on 25 August 2021).), which had a spatial resolution of 30 m and cloud-free, acquired on 8 January 2021, 27 April 2020, 14 July 2019 and 18 September 2020, to represent winter spring, summer, and autumn, respectively. The radiative transfer equation can be express as:(1)$$L_{\lambda }=[\varepsilon \;B(T_{s})+(1-\varepsilon )\;L_{↓}]\tau +L_{↑}$$
where *L_λ_* represents the thermal radiation intensity of the wavelength received by the satellite sensor, *ε* is the land surface emissivity (LSE), and B(T_s_) represents the radiation brightness received by the sensor for a blackbody with temperature (W·m^−2^·sr^−1^·µm^−1^). *τ* is the atmospheric transmissivity. L_↑_ and L_↓_ are the upwelling and downwelling atmospheric radiances. Atmospheric profile parameters (L_↑_, L_↓_, and τ) were obtained by NASA (http://atmcorr.gsfc.nasa.gov/(accessed on 28 August 2021)) [52].

First, ENVI 5.3 is used to preprocess the raw images, including radiometric calibration, atmospheric correction, cutting, splicing, etc. The LSE is required when using RET to retrieve surface temperature. Referring to the numerical value in the study of José A. Sobrino [55], the LSE can be obtained as follows:(2)$$\varepsilon =0.004\;P_{v}+0.986$$
(3)Pv=NDVI−NDVIsoilNDVIveg−NDVIsoil

In Equation (3), *NDVI_soil_*, and *NDVI_veg_*, correspond to the values of *NDVI* for bare land (*P_v_* = 0) and a surface with a fractional vegetation cover of 100%, respectively [56]. After processing LSE, according to Plank’s law, B(T_s_) can be expressed as:(4)BTS=Lλ−L↑−τ 1−εL↓τε
where *T_s_* is the real land surface temperature (*K*), then according to the inverse function of Planck’s formula, the real surface temperature *T_s_* can be obtained as follows:(5)TS=K2lnK1BTS+1
where *K*_1_ and *K*_2_ indicate the preset constant before the satellite launch, which is presented in the metadata MTL file query, for the Landsat-8 TIRS 10th, *K*_1_ = 774.8853 W m^−2^ sr^−1^ µm^−1^, *K*_2_ = 1321.0789.

#### 2.2.2. Remote Sensed Urban Park

We use high-spatial-resolution WorldView-2 satellite images based on landscape ecology theory and ENVI 5.3 combined with the human-computer interaction interpretation method of manual correction to divide the land cover types. In this study, supervised classification is applied using maximum likelihood, a parametric classification based on a Gaussian probability density second-order statistic model, based on WorldView-2 imagery at 0.5 m resolution on 23 December 2020. The land cover types of UP in the study area are classified into four types: vegetation (all green plants, including woodland, shrubland, and herbaceous), bare land (all areas containing exposed and non-developed surfaces, including sand, rocks, and soil), water bodies (including ponds, streams, and fountains), and impervious surfaces (including buildings, squares, and other facilities) (Figure 3). The training samples are optimized based on the auxiliary data, with subsequent manual visual modification to play the computer’s automatic pattern recognition function and the interpreters’ comprehensive judgment to correct the parts with significant differences in machine interpretation. The land cover classification was assessed for accuracy; the overall accuracy rate was 96.7363%, and the kappa coefficient was equal to 0.9726, satisfying the accuracy assessment requirements [57].

Based on the statistical yearbook of Luoyang and municipal planning and field surveys, the boundaries of UP in Luoyang were manually extracted based on high-resolution image maps. A total of 123 parks in the study area were extracted and classified using various classification features, with the four main criteria listed below. To begin with, the national industry standard “Standard for classification of urban green space” (CJJ/T85-2002) combined with the “Code for classification of urban land use and planning standards of development land” (GB50137-2011). Some of the street gardens are reclassified into pocket parks (PP) and plaza parks (PLP), which are finally classified into five categories: pocket parks (PP), community parks (CP), plaza parks (PLP), comprehensive parks (COP), and specific parks (SP). According to the nature of the parks, they are classified by area as mini parks (≤2 hm^2^, MP), small parks (2–5 hm^2^, SMP), medium parks (5–10 hm^2^, MMP), large parks (10–50 hm^2^, LP), and super large parks (≥50 hm^2^, SLP). Water is the most critical landscape type affecting the park’s LST and provides the most excellent cooling effect. Therefore, parks are divided into parks with water (WP) and without water (NWP). The land cover of parks is the major impact of LST; the four land cover types are divided according to the direction of impact. Water and vegetation can have a negative effect on the park’s LST; therefore, the parks where water bodies and vegetation account for more than half of the total area are defined as Vegetation&Water parks (VWP). Bare&Imper parks (BIP) are those that have more than half of the area covered by impervious surfaces and bare land. The remaining parks are classified as “balance parks” (BP) since no category of land cover makes up more than half of the total park area.

#### 2.2.3. Landscape Metrics

Numerous metrics have been developed to measure and describe landscape patterns. Landscape composition and configuration are two fundamental aspects of the system used to classify landscape patterns. Landscape composition is typically described by the landscape types and their proportions, whereas landscape configuration focuses on describing the spatial characteristics of individual regions and the spatial relationships among the multiple regions [58,59]. Based on empirical knowledge of the inherent features of landscape metrics and the nature of the practical study [59,60,61], a set of landscape metrics was selected to investigate the effect of landscape pattern on LST. Combined with the current situation of the study area, a total of 26 specific indicators are selected from three aspects: landscape composition, landscape configuration index, and park shape index, to quantify the park landscape pattern. (Appendix A) Out of them, the park shape index, the area, perimeter, shape index, etc., of the park are described, which characterize the overall shape of the park. Landscape composition represents the proportion and area of each land cover type in the park and characterizes the distribution differences of each landscape type. Landscape configuration, including those describing the number and density of patches inside the park: PD, NP, indicators describing patch morphology in parks: PAFRAC, PARA_MN, FRAC_MN, SHAPE_MN, AREA_MN, LSI, indicators relating patch distribution characteristics within the park: AI, COHESION, IJI, and indicators describing plaque diversity: SHEI, SHDI, and SIDI, together, represent the fragmentation, connectivity, and diversity of the landscape within the park. The above landscape pattern results were calculated using the formula combined with Fragstats 4.2.

#### 2.2.4. Correlation between LST and Landscape Pattern

As we all know, urban park LST is affected by many driving factors. What is less known, however, are the dominant factors [62]. Various statistical analysis methods were applied to identify the correlation between LST and its influencing factors. The Shapiro-Wilk test (>0.05) was applied to ensure that our sampled dataset follows the normal distribution. Afterwards, variance analysis and the independent sample T-test were used to analyze whether there were significant differences in the LST of different parks in different seasons. The average LST of parks in each season was taken as the dependent variable, and various parks were used as factors. This method identified whether there are significant differences in LST between different parks in the four seasons with the significance test at the 0.05 level. After that, Pearson correlation analysis (*p* < 0.05) was used to determine whether there was a statistically significant relationship between LST and landscape metrics in different seasons and determine the influence of various landscape metrics on LST. Furthermore, stepwise multiple linear regression was used to model the linkage between LST and landscape metrics. The best-fit model was selected based on regression statistics (R2, *p*-value), and its coefficients are statistically significant. Collinearity between the independent variables was evaluated regarding the variance inflation factor (VIF). The final regression model was obtained after removing the variables with collinearity from the model and re-analyzing them. Several analytic methodologies were integrated to determine the leading factors of LST and analyze the importance ranking of its influencing factors in order to propose better strategies to alleviate UHI.

## 3. Results

### 3.1. Spatial Heterogeneity of LST in Different Urban Parks and Different Seasons

The LST varied significantly in various parks and seasons (Figure 4 and Figure 5). The mean LST of all parks ranged from −3.90 °C to 1.78 °C in winter, from 22.16 °C to 30.32 °C in spring, from 30.85 °C to 38.60 °C in summer, and from 28.99 °C to 38.93 °C in autumn, respectively (Figure 5 and Table 1). The mean LST of all parks was 26.68, 35.39, and 34.09 °C in spring, summer, and autumn, and was lower by about 1.21, 1.15, and 1.22 °C than that of urban areas, respectively, while the mean LST of all parks in winter was −0.94 °C and higher at 0.89 °C than that of urban areas. In other words, parks can play a role in regulating surface temperature in all seasons. The descriptive statistical results of the LST in all parks and urban areas are shown in Table 1.

The classification of nature, area, water, and landscape pattern all significantly affected the LST of parks in all seasons except winter (Figure 6). The mean LST of the PP, CP, and PLP was significantly higher than that of the SP and COP (Figure 6e,i,m). The LST of NWP is substantially higher than WP (Figure 6d,l,p). In parks of various sizes, the MP, SMP, and MMP had a higher LST in spring, while the SLP had a considerably lower LST in summer and autumn (Figure 6f,j,n). What’s more, the LST of the GWP and BP was significantly lower than the BIP in the different landscape categories that dominate parks (Figure 6g,k,o).

### 3.2. Spatial Changes in Landscape Metrics

Same with LST, the distributions of landscape patterns had considerable variations in different categories of parks. The park area is distributed between 0.04 and 587.76 hm^2^, with the longest park reaching a perimeter of 9.27 km. In the parks, the land cover type with the largest average percentage was vegetation (54%), followed by impervious surfaces (28%), bare land (10%), and only 8% of water bodies.

The landscape metrics difference of various parks is shown in Figure 7. AREA, PERIM, Ci, Cv, COHESION, and NP were significantly lower in CP and PP than in COP, while the opposite was true for PD (Figure 7c,d,i,k,q,z,y). SIDI and AREA_MN were significantly larger in SP than in PP (Figure 7p,w). Pi in PLP is significantly larger than PP and SP, while the Pv of PP is significantly smaller than that in PLP (Figure 7f,g).

With an increased area in parks of various sizes, AREA, Ci, and Cv greatly grew (Figure 7c,i,k). SHDI and SIDI for LP were significantly larger than MMP, while the opposite is true for LSI (Figure 7o,p,x). PD was significantly larger in MP and SMP than that in SLP and COHESION, and PERIM was just the opposite (Figure 7y,q,d). SIDI, AREA_MN, and NP were significantly larger in LP than in MP (Figure 7p,w,z).

Among parks with different land cover, the SHDI, SIDI, PAFRAC, and SHEI of BP were significantly greater than those of other parks (Figure 7o,p,s,n). The Pv, Cv, and FRAC_MN values in BIP were significantly smaller, and Pi was conversely (Figure 7g,k,u,f). CIRCLE, SHAPE Pv, Cv, Cw, and AREA in VWP were significantly higher than in BIP (Figure 7a–c,g,k,l).

The Pi and PD of WP were significantly lower than NWP. Moreover, except for SHAPE_MN, PARA_MN, PAFRAC, IJI, Pb, Pv, SHAPE, and CIRCLE, all indicators of parks with water are significantly larger than those of parks without water.

### 3.3. Correlations between LST and Landscape Pattern

For all the landscape metrics examined in this study, the direction and magnitude of their impact on LST were generally different in different seasons (Figure 8). The Pearson correlations between LST and landscape patterns showed that the LST was negatively correlated with Pw, Pv, Cw, Cv, AI, AREA, and AREA_MN and positively correlated with Pb, Pi, and PD, respectively. The correlation between landscape pattern and LST was more effective in spring, summer, and autumn.

The LST of the PP and CP was positively correlated with Pb and PD and negatively significantly correlated with Cw, AI, and AREA_MN, respectively. The correlations between PLP LST and Pv were negative except for winter, and Pb was positive in summer and autumn. In spring and summer, the LST of COP positively correlated with Pi and AI, and Pv was the most substantial negative performer. Unlike other seasons, LST showed a strong negative correlation with PERIM (−0.99) and AREA (−0.94) in winter. The correlations of LST of SP to SHEI, SHDI, and SIDI were positive (0.71~0.87) and negative to Pv (−0.68~−0.8) and AI (−0.64~−0.79) in spring, summer, and autumn, respectively.

Among the parks of different sizes, the LST of the MP in winter was positively correlated with Cb (0.42) and Pb (0.41). Meanwhile, the LST of the SMP in spring was positively correlated with Pb (0.5) and PAFRAC (0.54). Except for winter, LST was positively correlated with Pw, Cw AI, and AREA_MN and negatively correlated with PD for LP, respectively. The landscape coefficients associated with the LST of SLP were higher, with significant positive correlations with SHEI, SHDI, and SIDI, significant negative correlations with Pv and AI, and the strongest negative correlation with Pb (−0.84), respectively.

Moreover, VWP LST was negatively correlated with AREA_MN, AI, Pw, and PD, respectively. BP LST was positively correlated with Pb (0.59) and negatively correlated with Pi (−0.57) in winter and AI (−0.58) and Pw (−0.76) in summer, respectively. In contrast, AREA_MN (−0.76) was negatively correlated with VWP in the summer. It was worth noting that the LST of WP significantly but inconsistently correlated with landscape patterns. For example, it was positively correlated with the diversity index in winter, negatively and strongly associated with AI and AREA_MN in other seasons, and negatively correlated with PD in all seasons. In spring, the LST of the NWP was negatively correlated with Pv (−0.48) and positively correlated with LSI (0.31) and Cb (0.33) in winter, respectively.

### 3.4. Relative Importance of Landscape Driving Forces

The stepwise regression analysis showed that park LST changes are closely related to landscape patterns, and the specific influencing factors and their degree of influence vary seasonally and by park type, with landscape patterns explaining significantly more of the variation in LST by season than park shape factors (Figure 9). Overall, the regression model explained the LST variation more in spring and less in winter. For the park categories, the landscape driving forces contributed more in the PLP and COP than other parks in all seasons (Figure 9a–d). The LST variation of the CP was mostly affected by AREA_MN in spring, summer, and autumn, and by Pb in winter. While the LST variation of the PLP was mostly affected by PERIM (82%), AI (74.5%), and Pi (92.5%) in spring, summer, and autumn, respectively. For the COP, the most important factor that impacted LST was PREIM in autumn and winter, Pv in spring, and Pi in summer, respectively. In comparison, PD and SHEI contributed more to the LST variation in summer and autumn, respectively.

Generally speaking, the larger the park area was, the better the fit of the regression model was, and the clearer the influencing factors of LST were (Figure 9e–h). Pb and Ci played an essential role in the MP, SMP, and LP in all seasons. In contrast, AREA_MN was the dominant driver for the LP in spring (82.3%), summer (75.0%), and autumn (84.5%). For SLP, the unique variables introduced in the regression models for winter, summer, and autumn were PARA_MN (40.3%), SHEI (45.5%), and Pb (62.3%), respectively.

Among the parks with different landscape percentages, the lower the percentage of water bodies and vegetation was, the lower the explanation of the regression model was (Figure 9i–l). In all seasons, the LST was most affected by Pb in the BP and AREA_MN in the VWP, respectively. It should be noted that the BIP was not affected by the landscape pattern. Moreover, the landscape driving forces contributed more to WP than NWP in all seasons except winter (Figure 9m–p). PD and AREA_MN were the dominant influencing factors in WP, while Pv was the most significant influencing factor in NWP.

## 4. Discussion

### 4.1. Seasonal Differences of LST in Park Green Space

Our result indicated that Luoyang’s overall thermoregulation intensity values are in the range of 1–2 °C, with the most robust cooling at 2.75 °C, which confirmed that the UP could mitigate the UHI effect. We also found that the mitigating effect of parks on UHI varies by season in Luoyang, being strongest in the autumn, followed by spring and summer, although the UP had a lifting influence on LST in the winter. In the intensity of the UP to regulate the urban thermal environment, our result is in agreement with previous studies that the UP is an essential urban landscape for thermal environment regulation [28,63,64]. In contrast, the cooling capacity of UP in the study area is poor compared to the results of Chen in Wuhan. Several complicated elements, including the background climate, urbanization level, and human activities, influence the capability of UP to regulate surface temperature [7,65,66]. Therefore, the results of the studies conducted in different cities differed somewhat. On seasonal differences, in contrast to our findings, the relevant studies indicate that the moderating effect of UP on UHI was greatest in summer and insignificant in winter. [32,67,68,69]. However, different results were found in studies in South Africa, and the mitigation effect was the greatest in spring and the least in autumn.

The relationship between UP and UHI depends on local climate conditions, the heterogeneity in ecological conditions of different climate zones contrasts sharply [70]. Generally, the cooling effect of UP becomes more potent when the background temperature is higher. [71,72]. They are related to solar radiation, the moisture content of the air, and rainfall, which affect the vegetation indirectly. The seasonal variation in solar height causes the ground to receive more solar radiation in the spring and summer and less in the winter. In winter, the combined effect leads to a weaker mitigating effect; the heat sources and emissions differ between winter and other seasons [68], and vegetation defoliation significantly increases the radiation transmission rate compared to other seasons [32,73]. Through biogeochemical and biogeophysical processes, precipitation can determine the seasonal fluctuations in LST [74]. Across the entire study period, Luoyang experienced a persistent drought in the summer, resulting in an extended absence of precipitation in the study area, which inhibited plant development, while the autumn weather remained wet. The precipitation was substantial and accompanied by torrential downpours. Thus, the condition of the vegetation was favorable. Hence, the UHI mitigation impact in autumn was greater than in summer. The spatial distribution of LST results indicates that parks on both banks of the Luo and Yi rivers have a lower LST than parks in other regions. Previous studies demonstrated that water bodies could form a local island. Due to the high specific heat capacity of water, increasing the temperature needs to absorb a lot of heat, limiting the rise in temperature to a certain extent and effectively reducing the LST in the surrounding areas, playing a positive role in mitigating the heat effect [75,76,77].

### 4.2. Interactive Effects of Seasonal Differences and Landscape Patterns on LST

In this study, landscape patterns were used to characterize the composition features of UP. Many studies have shown that the landscape pattern highly correlates with LST’s temporal variation and spatial distribution [65,78]. By comparing the main landscape factors influencing LST, we found that the driving factors of LST patterns in Luoyang city are complex, seasonal, and non-static. The result shows that the area and proportion of vegetation and water significantly affected LST in spring and summer but not in winter. In contrast, the cooling effect of water was more substantial than that of vegetation, which is similar to previous studies [41,70,74,79,80]. Vegetation mitigates UHI by having a high albedo, a low heat storage capacity, and reducing heat losses through evaporation and transpiration processes. It can also provide cooling through the shade formed by leaves to reduce the increased temperature of sun-exposed terrain [81,82,83]. Low temperatures in winter, diminished solar radiation, and UHI are mainly caused by human activities and thermal pollution, whereas vegetation and water bodies have a lesser effect on LST [84]. While some results from this study were anticipated, several findings merit further discussion. It has long been observed that increased Pi and Pb could increase LST [81,85,86,87]. However, our findings suggest that Pb had a broader and stronger influence and was the most significant driver of increased LST in all types of parks in winter, spring, and fall (Figure 8).

The positive effect of Pb on LST has been mentioned in many studies, which is due to heat concentration, low water-holding capabilities, minor thermal inertia, deficient relative humidity, and topsoil aridity in the upper layers of the sandy soil. Additionally, variations in LST in the bare areas may be due to the soil properties (color, crust, type of minerals) [88]. However, the degree of its impact on LST is weaker than that of impervious surfaces. Since the park construction in Luoyang City is still in the refinement stage, there is a large amount of bare ground in many parks. Compared with other cities, the proportion of bare land in parks is more pronounced, and as a consequence, bare land exhibits a more significant impact. It is of interest that the study results in BP do not support the general conclusion presented above, with a significant negative correlation between Pb and LST (Figure 8i,j,l). The negative correlation between Pb and LST suggests that an increase in surface area in BP may have a cooling effect, which contradicts our conventional view that bare land tends to increase urban temperatures. Does an increase in Pi in balance lead to a lower LST? Subsequent landscape type temperature calculations and stepwise regression analysis for BP showed that Pb did not show a decrease in LST (Table 2 and Figure 8i–l).

In addition to the landscape composition factor, AREA_MN had the most significant negative effect on LST in spring, summer, and autumn, and PD had the most significant positive impact on LST. PAFRAC had the most important positive effect on LST in the winter. AREA_MN and PD have been important landscape pattern factors in LST-related studies [89], which can characterize the degree of landscape fragmentation [90]. The significant negative contribution of AREA_MN to LST and the positive contribution of PD indicate that increased landscape fragmentation significantly exacerbates the UHI effect. This phenomenon is prevalent for most of the year, as has been generally reported and verified in previous studies [70,91,92,93]. The effects of patch area and density on LST can be traced back to the differential effects of different landscape types of patches on LST [94]. It is believed that AREA_MN of impervious surface patches provides a positive impact on the increase of LST. In contrast, the opposite is true for vegetated patches. Increased urbanization has led to the fragmentation of impervious surfaces and artificial green space patches, resulting in a fragmented urban landscape pattern, while reducing the degree of urban landscape fragmentation can reduce urban LST [95].In this study, PARFAC also had a significant positive effect on LST. However, this effect was only observed in winter and spring, which we believe is because the overall vegetation growth is poor during these seasons. Vegetation and other landscape elements have relatively little effect on the park during these seasons due to their lower temperatures; it is mainly the park shape that controls the internal heat transfer and affects LST. Previous studies have demonstrated that PARFAC regulates park LST by influencing the shape and area of the park and affecting the physical connection between parks and the surrounding area, thereby accelerating energy flow and conversion [48,96].

### 4.3. Implications for Urban Landscape Management

Our work shows the influence of landscape pattern on LST in different parks, the specific correlations, and the potential value of managing urban surface temperatures to reduce heat exposure risk. Landscape patterns can be controlled by design, and urban planners can optimize UP internal landscape patterns to regulate urban LST to explore optimal cooling patterns and enhance urban ecology. The fraction of each type of landscape is the most significant element influencing LST, as determined by the results of our study. Vegetation and water are beneficial in providing a cooling effect [28,49,97]. Tiny lakes and ponds play an unexpectedly important role in mitigating the urban heat island effect, which is similar to Chen’s study [98]. According to this, we propose to introduce water in waterless parks, if the introduction of live water is difficult, artificial reservoirs or ponds can be built. Parks with large waters can break them up into smaller, scattered ones, as suggested by Chen [98], to improve the park’s ability to reduce the LST.

The impervious surface and bare land produce the highest regional heat. Impervious surfaces are difficult to change extensively due to their nature and role. However, bare land can be reduced by enhancing vegetation planting. Increasing the quantity of vegetation is thus the principal focus of the park’s rehabilitation [99]. Reducing the proportion of impervious and bare land in the parks [50,100], bolstering the planting of vegetation on the bare surface, and making the sparse vegetation gradually dense through vegetation maintenance may be an effective way that is less noticed by previous studies.

The landscape pattern in different parks can also provide various degrees of influence on LST. As previous studies mentioned in the suggestions for urban management before, increasing perimeter, area [38,100], morphological complexity [15], landscape diversity, and plant community diversity within some parks can reduce surface temperature [47,89], which is the same as our proposal. It is worth noting that Our work summarized some particular transformation models for parks with low cooling impacts by restructuring the landscape, modifying the size and position of landscape types, adjusting the patch density and park layout, etc. The pattern diagram is recommended for various types of UP with high LST (Table 3) to promote the mitigation of UHI in UP in Luoyang accurately.

### 4.4. Limitations and Suggestions

This study selected 22 standard features describing landscape patterns and 4 park morphology indices to provide a more comprehensive description of the landscape patterns of UP in Luoyang City. The chosen variables explained 78.2% of the variation in surface temperature. However, the mechanism of landscape patterns on the LST of parks over various seasons requires additional research. Managing the cooling impact of UP by altering landscape patterns is an important study topic. While most research on the cooling effect of UP is measured using remote sensing images, this paper likewise employs remote sensing images of a single period to describe LST in different seasons, despite the difficulty of obtaining continuous, high-quality meteorological data. The results may deviate from the actual results in a practical sense. This study considered investigating LST during the daytime. Therefore, adding nighttime data, multiple daytime readings, and viewing the winter season could show interesting variations over time. Therefore, future research should focus more on using more accurate field survey data to supplement remote sensing image data and introducing multi-source data into the study, combining multiple quantitative methods to measure LST and its influencing factors accurately. Future research on landscape characteristics should include a wide range of geographically distributed cities. This research should be conducted in as many diverse cities as possible and focus on the investigation’s practical implications and effects.

## 5. Conclusions

Rapid urbanization has produced a profound heat island effect globally, which has caused great harm to the ecological environment and human health. The main factors affecting LST and the intensity of their products are different in different seasons. Still, their main factors are interrelated, so it is significant to understand the specific patterns of UP to mitigate the urban heat island effect for park green space design and urban planning. To assist planners in tackling climate change, heat waves, and general high urban temperatures, we seek to determine how different urban characteristics are associated with land surface temperatures. The results strongly support initiatives for increasing green infrastructure in cities. We found that the landscape pattern explained to a great extent the variation of LST and its regulation of UHI in the UP (78.2%), and the park had the worst ability to regulate LST in winter and the strongest in summer. Patch density and type determined the spatial variation of LST in the exploration area. Four different ground cover types significantly affected park LST in different seasons. In contrast, diversity factors and patch distribution had little effect on LST variation. The PD, Pb, and Pi had a negative impact on UHI mitigation, which was most prominent in the spring and summer. The unfavorable impact of PD, Pb, and Pi was more evident in the spring and summer. The influence of landscape variety and the form component of the park itself were much more pronounced in winter. With greater LST, the more complicated the park shape and the more diverse the landscape type. For the landscape pattern selection, 22 standard features describing the landscape pattern and 4 indices describing the park morphology were selected for this study, which explained the variation in surface temperature (78.2%). However, the process of changing landscape patterns on the LST of the park over the various seasons requires more research. Meanwhile, this study paid less attention to the internal composition of plants. We mainly focused on the overall percentage and area of vegetation, which can be combined with empirical measurements in future studies to introduce multi-source data to quantify the pattern changes of vegetation and climate in detail. According to the results, urban planning and managers can justify planning the park’s shape, increase the proportion of greenery and water bodies, replace impervious surfaces with bare land, and reduce the degree of landscape fragmentation, which can alleviate the urban heat island effect to a greater extent and provide a comfortable urban ecological living environment for urban residents.

## Figures and Tables

**Figure 1 ijerph-20-03155-f001:**
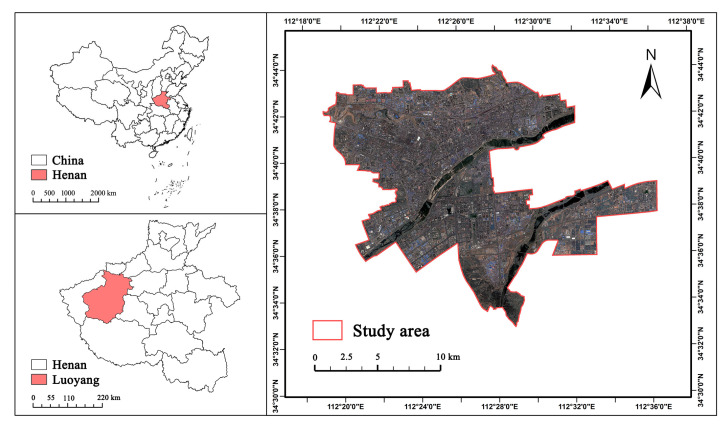
Location of Luoyang city, Henan, China.

**Figure 2 ijerph-20-03155-f002:**
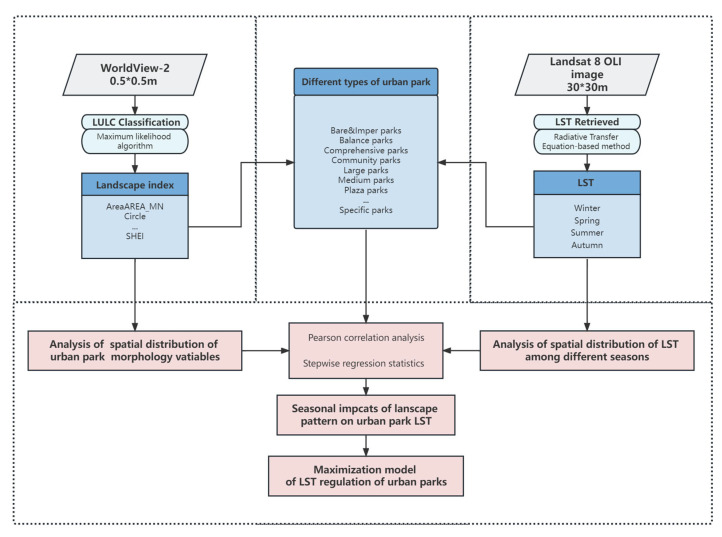
Flowchart of this study.

**Figure 3 ijerph-20-03155-f003:**
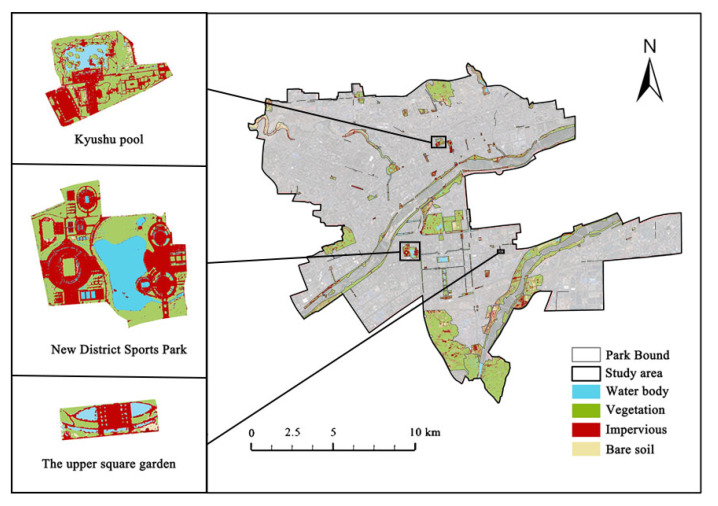
The land cover classification of Luoyang.

**Figure 4 ijerph-20-03155-f004:**
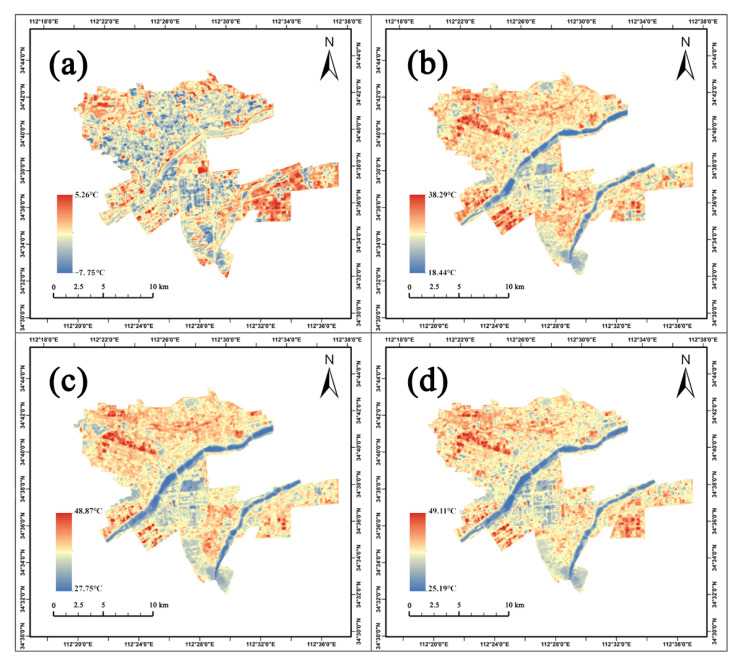
Spatial distribution of land surface temperature (LST). (**a**) LST in winter (**b**) LST in spring (**c**) LST in summer (**d**) LST in autumn.

**Figure 5 ijerph-20-03155-f005:**
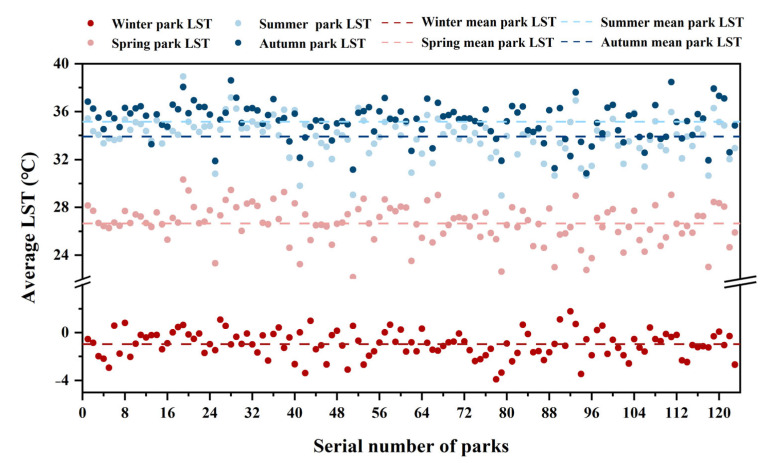
LST of parks in four seasons.

**Figure 6 ijerph-20-03155-f006:**
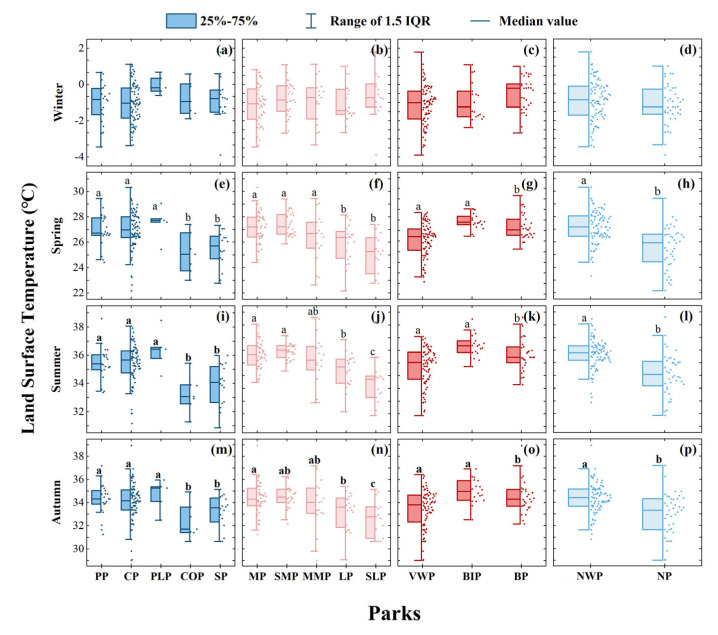
Park category differences of LST in different seasons (**a**–**p**). a, b, c, represent significant differences determined by Fisher’s least significant difference (LSD) tests (*p* < 0.05) on different seasons for different park types.

**Figure 7 ijerph-20-03155-f007:**
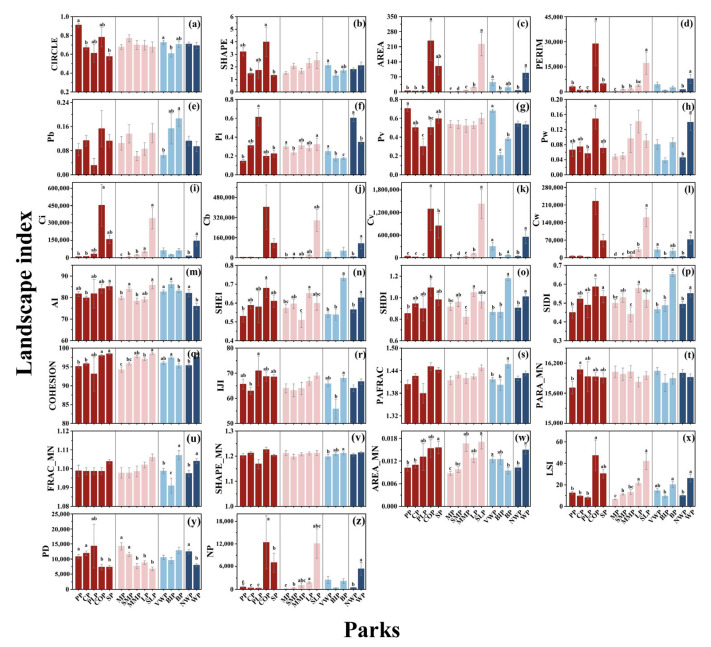
Landscape metrics differences in different categories of the park (**a**–**z**). a, b, c, d, e represents significant differences determined by Fisher’s least significant difference (LSD) tests (*p* < 0.05) on the same landscape metric for different park types.

**Figure 8 ijerph-20-03155-f008:**
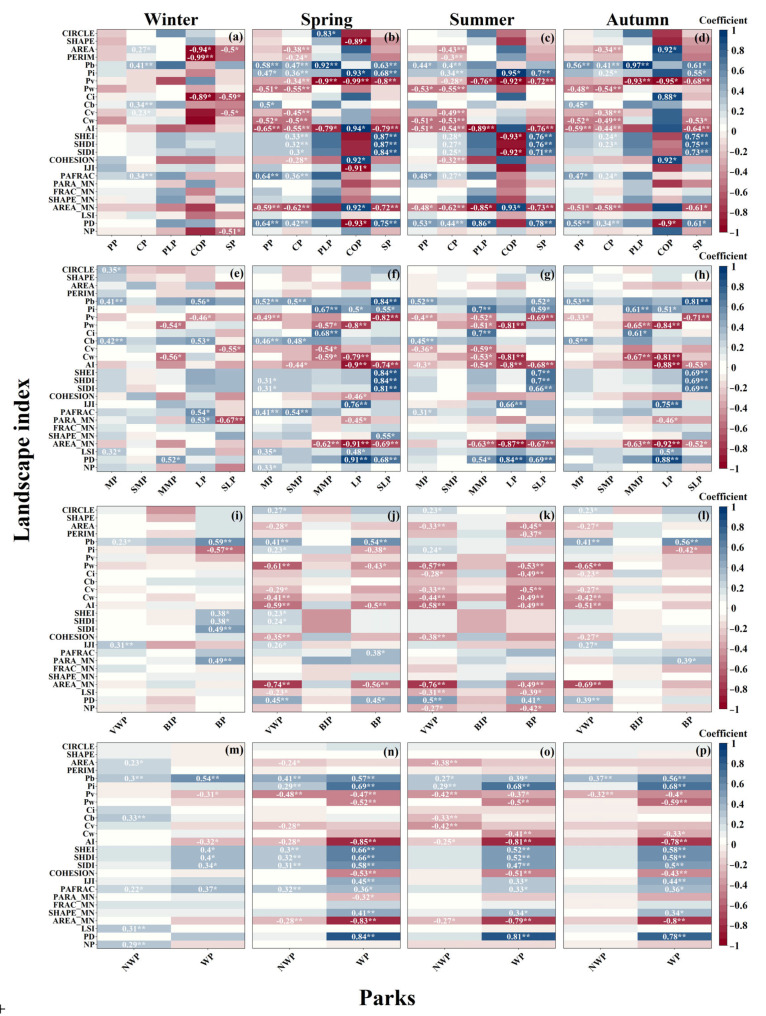
Heatmaps of correlation coefficients between landscape patterns and LST in different types of parks (**a**–**p**). The number of stars shows the significance level of the coefficient, ** *p* < 0.01, * *p* < 0.05.

**Figure 9 ijerph-20-03155-f009:**
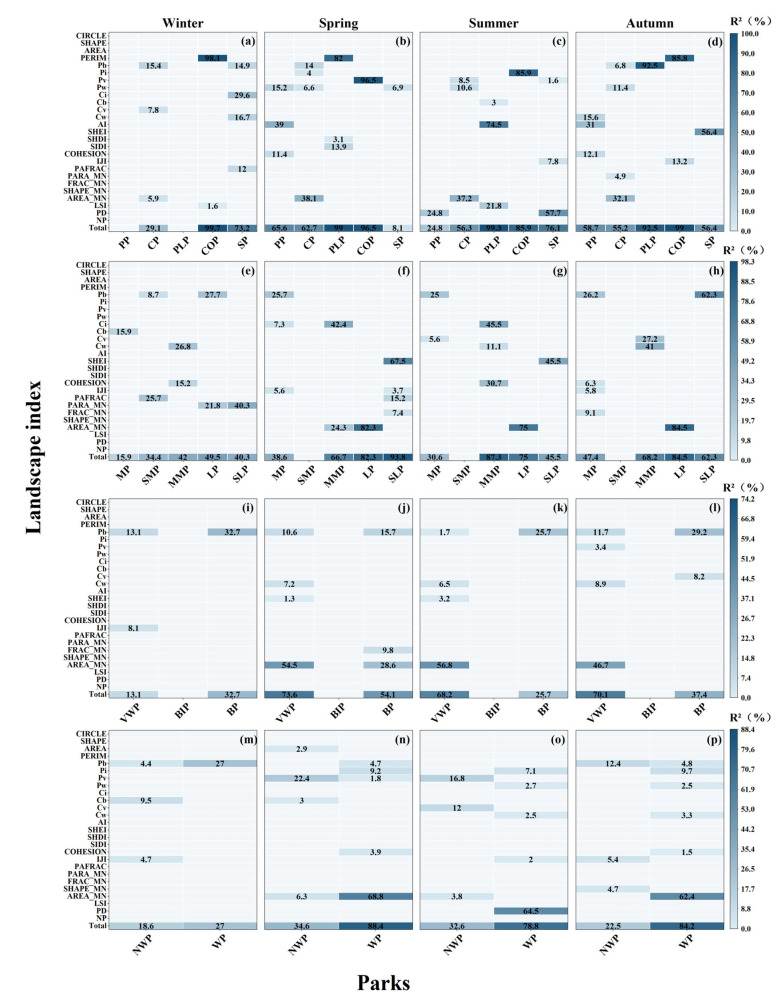
Stepwise regression diagram of landscape pattern and LST in different parks in four seasons (**a**–**p**).

**Table 1 ijerph-20-03155-t001:** Park LST and LST differences in different seasons.

Season	Park Green Space (°C)					The Temperature Difference between Green Space and Urban Area (°C)				
	Maximum	Minimum	Median	Mean	Standard Deviation	Maximum	Minimum	Median	Mean	Standard Deviation
Winter	1.78	−3.90	−0.94	−0.97	1.10	2.84	0.01	0.12	0.89	0.66
Spring	30.32	22.16	26.68	26.64	1.58	4.66	0.02	0.15	1.21	1.03
Summer	38.60	30.85	35.39	35.16	1.52	4.52	0.00	0.02	1.15	1.01
Autumn	38.93	28.99	34.09	33.92	1.64	5.13	0.01	0.04	1.22	1.12

**Table 2 ijerph-20-03155-t002:** LST of each landscape type in BP.

	Winter LST (°C)				Spring LST (°C)				Summer LST (°C)				Autumn LST (°C)			
Landscape Types	Minimum	Maximum	Mean	Standard Deviation	Minimum	Maximum	Mean	Standard Deviation	Minimum	Maximum	Mean	Standard Deviation	Minimum	Maximum	Mean	Standard Deviation
Water	−4.97	2.91	−0.39	28.98	19.31	31.19	25.71	2.25	28.98	39.95	33.74	1.97	26.57	39.46	33.05	2.34
Vegetation	−4.31	3.01	−0.24	29.69	20.75	32.16	26.50	1.51	29.69	40.07	34.80	1.60	28.21	39.71	34.02	1.59
Imperative	−4.86	3.12	0.39	30.63	22.77	32.14	27.63	1.47	30.63	40.58	34.49	1.71	30.83	39.18	34.88	1.31
Bare land	−5.04	3.04	−0.21	30.21	21.47	31.68	27.07	1.39	30.21	40.58	35.15	1.61	28.28	39.62	34.50	1.50

**Table 3 ijerph-20-03155-t003:** Patterns of the best and worst parks for LST regulation and the renewal of parks with poor effect on mitigating urban thermal environment.

Park Type	Views on the Transformation of Park Landscape Pattern	Pattern Diagram
All parks	Increase the amount of vegetation in the park, its area and proportion, and its density, creating low-maintenance rain gardens or sunken green areas. Increase the area of the park’s water features and plant floating and water-holding plants to enhance the purification function of water bodies and the surrounding environment. Reduce the area and proportion of the park’s impervious surfaces and bare land; strengthen vegetation maintenance on bare land and sparsely vegetated areas; and replace impervious surfaces with permeable paving or ecological paving. Increase the diversity of landscape types and species.	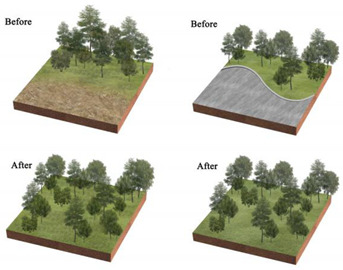
BP	Within the limited planting space, plant community structure and species types are enriched, and trees and shrubs are added to the lawn space to increase the number of species. The arrangement of vegetation patches in the park’s interior will be clustered as much as possible, and the average area of landscape patches will be increased. In contrast, the connectivity of landscape patches will be reduced. To maximize the size and simplify the park’s shape within a fixed block layout.	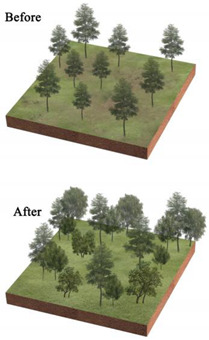
PLP	Focus on spatial coordination and balance, improving the road structure, and using permeable concrete and permeable tiles, other permeable paving in the plaza and road parts to ensure that the site runoff is organized to collect and infiltrate downward. Incorporate small plots around the park into the park, increasing the park perimeter and area and enhancing the complexity of the park’s shape from the morphological features. Various native trees and shrubs are added to the large lawn and bare land areas around the square to enrich the type of plant community structure and landscape richness while increasing the aesthetic value of the vegetation area.	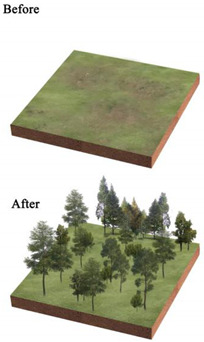
CP	Focus on the collocation of vegetation communities, improve the richness of plant communities, and increase the variety and collocation of plants in the limited greening area. Integrating the scattered landscape patches allows vegetation planting areas and impervious surface areas to be concentrated to enhance the aggregation of park patches and increase the average size of landscape patches. Improve the park road network, install environmental protection trails, use grass tiles and porous concrete to replace traditional trail materials, and use trail outreach to enrich the park shape and increase the park perimeter.	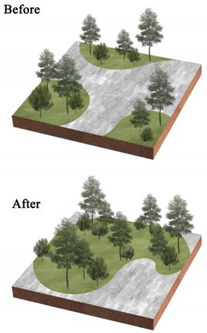
NWP	Pay attention to improving impervious surfaces, such as upgrading pedestrian walkways to permeable materials or materials with greater ecological benefits, or by upgrading traditional parking lots to ecological parking lots by installing percolation ponds and grass swales. Gathering infrastructure management facilities, buildings, and large plazas to reduce the density of patches and increase the average area of patches. Increasing the size of the park.	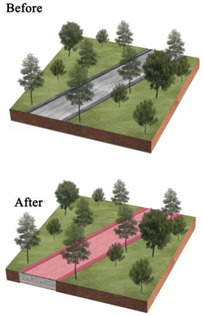

## Data Availability

Not applicable.

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
