# Peer review of "Discover the Desirable Landscape Structure of Urban Parks for Mitigating Urban Heat: A High Spatial Resolution Study Using a Forest City, Luoyang, China as a Lens"

_ijerph, 2023, doi:10.3390/ijerph20043155_

Round 1

Reviewer 1 Report

Using high resolution image data (WorldView-2), the manuscript titled “Discover the desirable landscape structure of urban parks for mitigating urban heat: a high spatial resolution study using a forest city, Luoyang, China as a lens” clearly identified land cover types of 123 parks in Luoyang using WorldView-2 data and selected 26 landscape indicators to quantify the park landscape characteristics, and analyzed the variations of the average land surface temperatures (LST) of 123 parks of various dimensions and structures in the Chinese city of Luoyang across the four seasons. This paper is interesting, the flow of the content is simple and clear, the methods are reasonably appropriate, and this study puts forward specific models and suggestions to assist urban planning.

Below, some specific issues to address are identified and revisions are suggested to improve the manuscript.

1)      Line 22: It is sugguested to change "high-precision data" to "high resolution data".

2)      UPS is recommended to be modified to UP.

3)      Line 44: The reference format needs to be modified.

4)      Line 88: It is sugguested to change “the LST of parkland” to “the park LST”

5)      Line 158-159: Add the references.

6)      Section 2.5: It is sugguested to change "Data processing" to "Correlation between LST and landscape pattern".

7)      The image precision needs to be improved to enhance the readability of the picture (such as Figure 6-8), and they should be enlarged or modified.

8)      Line 340-345 the number “-.099” is incorrect, the number in the result need to check.

9)      Line 482: The reference format needs to be modified, please check and correct it carefully.

10)   Line 493: Change “surface temperatures” to “LST”. The writing of Land surface temperature (LST)in the text should be uniform.

11)  Line 501: It is sugguested to change “Water bodies” to “Water”.

12) Some relavent references are recommended here:

Chen, M., et al. (2022). "Quantification and mapping cooling effect and its accessibility of urban parks in an extreme heat event in a megacity." Journal of Cleaner Production 334.

Chen, M., et al. (2023). "Carbon saving potential of urban parks due to heat mitigation in Yangtze River Economic Belt." Journal of Cleaner Production 385.

Gao, Z., et al. (2022). "Toward park design optimization to mitigate the urban heat Island: Assessment of the cooling effect in five US cities." Sustainable Cities and Society 81.

Author Response

Dear reviewer,

Thank you for your insightful comments concerning our manuscript entitled “Discover the desirable landscape structure of urban parks for mitigating urban heat: a high spatial resolution study using a forest city, Luoyang, China as a lens”. Those comments are all valuable and very helpful for revising and improving our paper, as well as the important guiding significance to our research. We have studied the comments carefully and have made correction which we hope meet with approval. Revised portions are marked in red in the paper. The main corrections in the paper and the responses to your comments are as flowing:

[Comment #Line 22]: It is suggested to change "high-precision data" to "high resolution data".

[Response # Line 22]: Thank you so much for your careful check. Apologies for our inappropriate expressions. We have corrected this in the manuscript (Line 22, 164 and 252) and also attached the revised version below.

Revised version:

The study's primary purpose is to investigate the relationship between LST and landscape features of different park categories based on high resolution data.

[Comment #1]: UPS is recommended to be modified to UP.

[Response # 1] Thanks very much for your comment. We have revised the abbreviations of all parts of the article. Thank you again for your suggestions.

[Comment #Line 44]: The reference format needs to be modified.

[Response # Line 44]: Thanks for your carefully check. Apologize for our format error. We have checked the reference format of the full manuscript and corrected them. Thank you again for your check.

[Comment #Line 88]: It is suggested to change “the LST of parkland” to “the park LST”

[Response # Line 88]: Thank you so much for your carefully check. Apologies for our inappropriate expressions. We have corrected this in the manuscript (Line 106-108)and also attached the revised version below.

Revised version:

The study of cities in different climate zones shows that the UP LST LST of parkland varies significantly across seasons and day and night[21,31,32].

[Comment #Line 158-159]: Add the references.

[Response # Line 158-159]: Thank you so much for your suggestions. We have added relevant references to support this sentence. Thank you again for your carefully check.

References:

José A. Sobrino; Juan C. Jiménez-Muoz; Leonardo Paolini Land Surface Temperature Retrieval from LANDSAT TM 5. Remote Sensing of Environment 2004, doi:10.1016/j.rse.2004.02.003.

[Comment # 2]: Section 2.5: It is suggested to change "Data processing" to "Correlation between LST and landscape pattern".

[Response # 2]:Thank you very much for your suggestions. We have changed the "Data processing" to "Correlation between LST and landscape pattern".

[Comment # 3]: The image precision needs to be improved to enhance the readability of the picture (such as Figure 6-8), and they should be enlarged or modified.

[Response # 3]:Thank you so much for your check .We have replaced the Figure 6-8 with clearer ones. And increase the font size of the Figure7-8 to increase the readability of the Figure. Thank you again for your suggestions.

[Comment #Line 340-345]: Line 340-345 the number “-.099” is incorrect, the number in the result need to check.

[Response # Line 340-345]:Thanks so much for your carefully check. Apologies for our clerical error. We have corrected this in the manuscript (Line 386) and also attached the revised version below.

Revised version:

Unlike other seasons, LST showed a strong negative correlation with PERIM (-0.99) and AREA (-0.94) in winter.

[Comment #Line 482]: Line 482: The reference format needs to be modified, please check and correct it carefully.

[Response # Line 482]: Thanks very much for your carefully check. Apologize for our format error. We have checked the reference format of the full text and corrected the error. Thank you again for your carefully check.

[Comment #Line 493]: Line 493: Change “surface temperatures” to “LST”. The writing of Land surface temperature (LST)in the text should be uniform.

[Response # Line 493]: Thanks very much for your suggestions and we are so sorry for the miswrite. We have checked and unified the abbreviations of all parts of the manuscript also attached the revised version below.

Revised version:

Our work shows the influence of landscape pattern on LST in different parks, the specific correlations, and the potential value of managing urban surface temperatures to reduce heat exposure risk..

[Comment #Line 501]: Line 501: It is suggested to change “Water bodies” to “Water”.

[Response # Line 501]: Thanks for your insightful suggestions. According to the comments of other reviewers, we have revised the manuscript in this part. The change you mentioned was resolved after modification, we have replaced "water bodies" with "water". Thanks again for your comment.

[Comment #4]: Some relevant references are recommended here:

Chen, M., et al. (2022). "Quantification and mapping cooling effect and its accessibility of urban parks in an extreme heat event in a megacity." Journal of Cleaner Production 334.

Chen, M., et al. (2023). "Carbon saving potential of urban parks due to heat mitigation in Yangtze River Economic Belt." Journal of Cleaner Production 385.

Gao, Z., et al. (2022). "Toward park design optimization to mitigate the urban heat Island: Assessment of the cooling effect in five US cities." Sustainable Cities and Society 81.

[Response # 4]: Thank you very much for your insightful suggestions. The references that were recommended are very relevant and so helpful for our manuscript, they provide some new ideas for us. Thanks again for your recommendation. We have read the references and added them to the appropriate place.

Special thanks to you for your good comments.

Reviewer 2 Report

Based on high spatial resolution images, the author examines the effect of urban parks on urban heat island, and discusses the relationship between different landscape structures of parks and land surface temperature, which is supplemented by a case study in Central China. The thesis of this paper is interesting and valuable. 

I. Introduction Section:

1.The introduction has not clearly demonstrated the research highlights of this paper. For example, how is this study different from previous studies? What's new? It is recommended to summarize this explicitly in the last paragraph of the introduction. 

2.There are many abbreviations of professional terms in the article. It is suggested to add "Abbreviations" after the abstract and keywords, and make a summary in the form of a table to facilitate readers to understand the article. 

3.It is also necessary to make the introduction more scientific. The factors affecting urban heat island and land surface temperature are still unclear, so the literature summary here needs to be added.  

II. Methodology Section:

1. Lines 161-165 of the method part mentioned that satellite images on January 8, April 27, July 14 and September 18 were selected to represent winter, spring, summer and autumn respectively. However, this paper does not mention why these four days are chosen as the representatives of the four seasons. Is it because the climate in these days is the most consistent with the characteristics of the four seasons? For example, is the date around January 8 the coldest of winter 2020? Please include the reason. 

2.As mentioned in line 207-210 of the method part, parks are classified according to the combination of the national industrial standard "Standard for classification of urban green space" (CJJ/ T85-2002) and "Code for classification of urban land use and planning standards of development land" (GB50137-2011). However, the Ministry of Housing and Urban-Rural Development of China has updated the "Standard for classification of urban green space" (CJJ/ T85-2002) to the "Standard for classification of urban green space" (CJJ/ T85-2017). Why not combine the new standard to classify parks? 

3.Many methods are used in this study. Before introducing the research design, a general description of the overall methodological process and a flow chart should be provided to help readers understand the whole process. 

4.Lines 234-237 of the method mentioned the summary of specific indicators from three aspects: landscape composition, landscape configuration index and park shape index. Why expand on these three aspects? What is the basis and source of all the indicators extracted? What are the indicators of innovation? 

5.Appendix A summarizes the surface temperature (LST) impact factors used in this study, but all from the same literature. Are these indicators recognized elements? 

III. Discussion and Conclusion Section:

1. In the discussion section, lines 492-518 discuss the implications for urban landscape management. However, this part has not been compared with previous studies. That is, according to others' research conclusions, which landscape structure or park element has a greater impact on urban heat island? What are the recommended design and management practices? Is this consistent with the findings of this paper? If there is a bias, what accounts for the difference?

Author Response

Response to the Review Comments

Dear reviewer,

Thank you for your insightful comments concerning our manuscript entitled “Discover the desirable landscape structure of urban parks for mitigating urban heat: a high spatial resolution study using a forest city, Luoyang, China as a lens”. Those comments are all valuable and very helpful for revising and improving our paper, as well as the important guiding significance to our research. We have studied the comments carefully and have made correction which we hope meet with approval. Revised portions are marked in red in the paper. The main corrections in the paper and the responses to your comments are as flowing:

[Comment #1]: The introduction has not clearly demonstrated the research highlights of this paper. For example, how is this study different from previous studies? What's new? It is recommended to summarize this explicitly in the last paragraph of the introduction.

[Response # 1]: Thank you for your insightful comments.

Our manuscript focuses on quantitatively the driving factors of land surface temperature of park green space in Luoyang, based on high resolution image. The base of the theory provides concrete suggestions for city management. We have revised the introduction and added relevant content describing the research highlight at the end of the introduction (Line 160-170) and also attached the revised version below. Thanks again for your suggestions.

Revised version:

Therefore, this study conducted an accurate definition of the landscape pattern base on the high-precision remote sensing data and a comprehensive comparison of the seasonal variation on the whole year of the effects of landscape pattern on park LST by specific park categories to puts forward the concrete and implementable urban planning scheme for urban management with the focus on (1) Use high resolution remote sensing data to obtain UPS landscape types and their shape indicators accurately. (2)Retrieve LST from remote sensing satellite infrared data, and probe the relationship between LST and park morphological indicators and landscape patterns. (3)Analyze the maximization mode of LST regulation of Luoyang Parks, and provide targeted recommendations and models for urban planning on how to mitigate the UHI effects through the rational allocation of the interior landscape of the park.

[Comment #2]: There are many abbreviations of professional terms in the article. It is suggested to add "Abbreviations" after the abstract and keywords, and make a summary in the form of a table to facilitate readers to understand the article.

[Response # 2]: Thanks very much for your suggestion. We apologize for the neglect of detail. We have added the part of "Abbreviations" after the abstract and keywords, and made a summary in the form of a table to facilitate readers to understand the article. Among them, the summary of abbreviations related to landscape pattern indicators is in Appendix A, so it does not appear in the abbreviation table. We attached the revised version below. Thanks again for your suggestions.

Revised version:

Abbreviations

BIP

Bare&Imper parks

PLP

Plaza parks

BP

Balance parks

PP

Pocket parks

COP

Comprehensive parks

R2

Regression statistics

CP

Community parks

SLP

Super large parks

LP

Large parks

SMP

Small parks

LST

Land surface temperature

SP

Specific parks

LULC

Land use/land cover

UHI

Urban heat island

MMP

Medium parks

UP

Urban parks

MP

Mini parks

VWP

Vegetation&Water parks

NWP

Parks without water

WP

Parks with water

[Comment #3]: It is also necessary to make the introduction more scientific. The factors affecting urban heat island and land surface temperature are still unclear, so the literature summary here needs to be added.

[Response # 3]: Thank you for pointing out the limitation of our work. We have revised the introduction of the manuscript and added the literature summary of the factors affecting urban heat island and land surface temperature(Line 37-53, 103-120) to make the introduction more scientific.

[Comment #Lines 161-165]: Lines 161-165 of the method part mentioned that satellite images on January 8, April 27, July 14 and September 18 were selected to represent winter, spring, summer and autumn respectively. However, this paper does not mention why these four days are chosen as the representatives of the four seasons. Is it because the climate in these days is the most consistent with the characteristics of the four seasons? For example, is the date around January 8 the coldest of winter 2020? Please include the reason.

[Response # Line 161-165]:The chosen of the dates are based on the division of seasons and the quality of the Landsat images. Firstly, we divided the LST into four seasons using three months as nodes according to the solar term method and the local temperature in Zhengzhou. The March to May, June to August, September to November, and December to February are divided into spring, summer, autumn, and winter respectively. Then the representative months of each season were selected according to the temperature, and the images which consistent with the characteristics of the four seasons and with the least cloud cover in the study area were selected for analysis. What is noted here is that the screen of the images was based on all available images in a year. I hope the explanation is helpful for you to understand the manuscript.

[Comment #line 207-210]: As mentioned in line 207-210 of the method part, parks are classified according to the combination of the national industrial standard "Standard for classification of urban green space" (CJJ/ T85-2002) and "Code for classification of urban land use and planning standards of development land" (GB50137-2011). However, the Ministry of Housing and Urban-Rural Development of China has updated the "Standard for classification of urban green space" (CJJ/ T85-2002) to the "Standard for classification of urban green space" (CJJ/ T85-2017). Why not combine the new standard to classify parks?

[Response # Line 207--210]: Thanks for your carefully check. This part is our misrepresentation. We referred to the "Standard for classification of urban green space" (CJJ/ T85-2017) but mistakenly wrote it as the "Standard for classification of urban green space" (CJJ/ T85-2002).

In the "Standard for classification of urban green space" (CJJ/ T85-2002) the parks are classified as Comprehensive park(G11), Community park(G12), Special park(G13), Strip park (G14), and the green space by the street(G15). In the "Standard for classification of urban green space" (CJJ/ T85-2017), the parks are classified as Comprehensive park(G11), Community park(G12), Special park(G13) , Strip park (G14), and the pocket parks(G15). Field investigation and remote sensing surface classification found that the impervious surface area of some parks is quite large, which is not in line with the park's standards but accords with the characteristics of the plaza, so they are classified as plaza parks.

We apologized for our misrepresentation and thanks again for your carefully check. We have corrected that in the manuscript(Line 251-256) and attached the revised version below.

Revised version: To begin with, the national industry standard "Standard for classification of urban green space" (CJJ/T85-2017) combined with the "Code for classification of urban land use and planning standards of development land" (GB50137-2011). Some of the street gardens are reclassified into pocket parks (PP) and plaza parks (PLP), which are finally classified into five categories: pocket parks (PP), community parks (CP), plaza parks (PLP), comprehensive parks (COP) and specific parks (SP).

[Comment #4]: Many methods are used in this study. Before introducing the research design, a general description of the overall methodological process and a flow chart should be provided to help readers understand the whole process.

[Response # 4]: Thanks for your insightful suggestion. We have created a general description on (Line 187-193) and a flowchart of the overall methodological process in the manuscript to summarize the method and attached the revised version below.

Revised version:

The flowchart of the study is as follows (Figure 2). First, the LST of the parks was retrieved from Landsat 8 OLI images, and parks were divided into four major categories and 17 sub-categories. Second, the LULC of the study area was classified based on the WorldView-2 images, and the landscape indicators were calculated. Based on the above data, analyze the spatial pattern and the LST of UP in different seasons, and then explore the impact trend of landscape pattern on the LST through correlation analysis and stepwise regression analysis.

[Comment #Line 234-237]: Lines 234-237 of the method mentioned the summary of specific indicators from three aspects: landscape composition, landscape configuration index and park shape index. Why expand on these three aspects? What is the basis and source of all the indicators extracted? What are the indicators of innovation?

[Response # Line 234-237]: The main purpose of this classification is to introduce landscape pattern index more clearly. These three aspects are mainly used to classify and describe the landscape pattern. Each part can represent a large aspect of the landscape pattern, while the index contained describes some broad characteristics of the landscape pattern, and each index represents a different significance.

The extraction of indicators is mainly based on the reference of some relevant researches on landscape patterns and LST and we adopted the indicators mentioned in the previous studies. There was not much innovation in indicators. On the one hand in our text study, we put all the indicators into the correlation analysis. A lot of them didn't show a correlation with LST and so we screened them out. On the other hand, we believe that some of the landscape indexes that did not appear in our manuscript are similar in meaning to the indexes we selected. And they were not mentioned in previous studies, which may be because some studies found that they were not related to LST. Therefore, we mainly chose the landscape indexes that appeared in previous studies. The references for selecting landscape pattern index are as follows. Hope my respond is helpful to you.

  1. An, H.; Cai, H.; Xu, X.; Qiao, Z.; Han, D. Impacts of Urban Green Space on Land Surface Temperature from Urban Block Perspectives. Remote Sens. 2022, 14, 4580, doi:10.3390/rs14184580.
  2. Han, D.; Yang, X.; Cai, H.; Xu, X. Impacts of Neighboring Buildings on the Cold Island Effect of Central Parks: A Case Study of Beijing, China. Sustainability 2020, 12, 9499, doi:10.3390/su12229499.
  3. Huang, J.; Wang, Y. Cooling Intensity of Hybrid Landscapes in a Metropolitan Area: Relative Contribution and Marginal Effect. Sustainable Cities and Society 2022, 79, 103725, doi:10.1016/j.scs.2022.103725.
  4. Li, X.; Zhou, W.; Ouyang, Z.; Xu, W.; Zheng, H. Spatial Pattern of Greenspace Affects Land Surface Temperature: Evidence from the Heavily Urbanized Beijing Metropolitan Area, China. Landscape Ecol 2012, 27, 887–898, doi:10.1007/s10980-012-9731-6.
  5. Li, X.; Zhou, W.; Ouyang, Z.; Xu, W.; Zheng, H. Spatial Pattern of Greenspace Affects Land Surface Temperature: Evidence from the Heavily Urbanized Beijing Metropolitan Area, China. Landsc. Ecol. 2012, 27, 887–898, doi:10.1007/s10980-012-9731-6.
  6. Maimaitiyiming, M.; Ghulam, A.; Tiyip, T.; Pla, F.; Latorre-Carmona, P.; Halik, Ü.; Sawut, M.; Caetano, M. Effects of Green Space Spatial Pattern on Land Surface Temperature: Implications for Sustainable Urban Planning and Climate Change Adaptation. ISPRS Journal of Photogrammetry and Remote Sensing 2014, 89, 59–66, doi:10.1016/j.isprsjprs.2013.12.010.
  7. Rakoto, P.Y.; Deilami, K.; Hurley, J.; Amati, M. Revisiting the Cooling Effects of Urban Greening: Planning Implications of Vegetation Types and Spatial Configuration. Urban For. Urban Green. 2021, 64, 127266, doi:10.1016/j.ufug.2021.127266.
  8. Wang, X.; Cheng, H.; Xi, J.; Yang, G.; Zhao, Y. Relationship between Park Composition, Vegetation Characteristics and Cool Island Effect. Sustainability 2018, 10, 587, doi:10.3390/su10030587.
  9. Weng, Q.; Liu, H.; Lu, D. Assessing the Effects of Land Use and Land Cover Patterns on Thermal Conditions Using Landscape Metrics in City of Indianapolis, United States. Urban Ecosyst 2007, 10, 203–219, doi:10.1007/s11252-007-0020-0.
  10. Wu, Q.; Tan, J.; Guo, F.; Li, H.; Chen, S. Multi-Scale Relationship between Land Surface Temperature and Landscape Pattern Based on Wavelet Coherence: The Case of Metropolitan Beijing, China. Remote Sens. 2019, 11, 3021, doi:10.3390/rs11243021.
  11. Xiang, Y.; Ye, Y.; Peng, C.; Teng, M.; Zhou, Z. Seasonal Variations for Combined Effects of Landscape Metrics on Land Surface Temperature (LST) and Aerosol Optical Depth (AOD). Ecological Indicators 2022, 138, 108810, doi:10.1016/j.ecolind.2022.108810.
  12. Xiao, R.; Cao, W.; Liu, Y.; Lu, B. The Impacts of Landscape Patterns Spatio-Temporal Changes on Land Surface Temperature from a Multi-Scale Perspective: A Case Study of the Yangtze River Delta. Science of The Total Environment 2022, 821, 153381, doi:10.1016/j.scitotenv.2022.153381.
  13. Yang, C.; He, X.; Wang, R.; Yan, F.; Yu, L.; Bu, K.; Yang, J.; Chang, L.; Zhang, S. The Effect of Urban Green Spaces on the Urban Thermal Environment and Its Seasonal Variations. Forests 2017, 8, 153, doi:10.3390/f8050153.
  14. Yin, J.; Wu, X.; Shen, M.; Zhang, X.; Zhu, C.; Xiang, H.; Shi, C.; Guo, Z.; Li, C. Impact of Urban Greenspace Spatial Pattern on Land Surface Temperature: A Case Study in Beijing Metropolitan Area, China. Landsc. Ecol. 2019, 34, 2949–2961, doi:10.1007/s10980-019-00932-6.

[Comment #5]: Appendix A summarizes the surface temperature (LST) impact factors used in this study, but all from the same literature. Are these indicators recognized elements?

[Response # 5]: The references listed in Appendix A mainly refer to the description and calculation of the landscape index. The landscape indices selected in this paper come from a wide range of sources mentioned in comment 4, most of them are referenced in our manuscripts. In addition to the previously widely used landscape indices in the references, some less-used landscape indices are also introduced to quantify landscape patterns. Hope my respond is helpful to you.

[Comment #6]: In the discussion section, lines 492-518 discuss the implications for urban landscape management. However, this part has not been compared with previous studies. That is, according to others' research conclusions, which landscape structure or park element has a greater impact on urban heat island? What are the recommended design and management practices? Is this consistent with the findings of this paper? If there is a bias, what accounts for the difference?

[Response # 6]: Thanks for your insightful suggestions. We have revised the discussion section based on previous studies, adding comparisons and analysis of differences with previous papers(Line 567-594). As we know, lots of previous studies discussing LST in relation to urban landscape management focus on the results while the specific measures for the management of the urban landscape are more general. The suggestions for urban management are elaborated in our manuscript in the form of a table (Table 3) to provide concrete and implementable recommendations for urban management, we attached the revised version below.

Revised version:

Our work shows the influence of landscape pattern on LST in different parks, the specific correlations, and the potential value of managing urban surface temperatures to reduce heat exposure risk. Landscape patterns can be controlled by design, and urban planners can optimize UP internal landscape patterns to regulate urban LST to explore optimal cooling patterns and enhance urban ecology. The fraction of landscape type is the most significant element influencing LST, as determined by the results of our study. Vegetation and water are beneficial in providing cooling effect[1–3], tiny lakes and ponds play an unexpectedly important role in mitigating the urban heat island effect which is similar to Chens study[4]. According to this, we propose to introduce water in waterless parks, if the introduction of live water is difficult, artificial reservoirs or ponds can be built. Parks with large waters can break up waters into smaller, scattered ones, as suggested by Chen[4] to improve the park's ability to reduce the LST.

 The impervious surface and bare land produce the highest regional heat. Impervious surfaces are difficult to change extensively due to their nature and role. However, bare land can be reduced by enhancing vegetation planting. Increasing the quantity of vegetation is thus the principal focus of the park's rehabilitation.[5].Reducing the proportion of impervious and bare land in the parks[6,7], bolstering the planting of vegetation on the bare surface, and making the sparse vegetation gradually dense through vegetation maintenance which may be an effective way that is less noticed by previous studies.

Landscape pattern in different parks can also provide various degrees of influence on LST. As previous studies mentioned in the suggestions for urban management before, increasing perimeter, area[7,8], morphological complexity [9], landscape diversity, and plant community diversity within some parks can reduce surface temperature [10,11], which is the same as our proposal. It is worth noting that Our work summarized some particular transformation models for parks with low cooling impact by restructuring the landscape, modifying the size and position of landscape types, adjusting the patch density and park layout, etc. The pattern diagram is recommended for various types of UP with high LST (Table 3) to promote the mitigation of UHI in UP in Luoyang accurately.

  1. Sun, R.; Chen, L. Effects of Green Space Dynamics on Urban Heat Islands: Mitigation and Diversification. Ecosystem Services 2017, 23, 38–46, doi:10.1016/j.ecoser.2016.11.011.
  2. Kuang, W.; Liu, Y.; Dou, Y.; Chi, W.; Chen, G.; Gao, C.; Yang, T.; Liu, J.; Zhang, R. What Are Hot and What Are Not in an Urban Landscape: Quantifying and Explaining the Land Surface Temperature Pattern in Beijing, China. Landscape Ecology 2015, 30, 357–373, doi:10.1007/s10980-014-0128-6.
  3. Li, X.; Zhou, W.; Ouyang, Z.; Xu, W.; Zheng, H. Spatial Pattern of Greenspace Affects Land Surface Temperature: Evidence from the Heavily Urbanized Beijing Metropolitan Area, China. Landscape Ecol 2012, 27, 887–898, doi:10.1007/s10980-012-9731-6.
  4. Sun, R.; Chen, L. How Can Urban Water Bodies Be Designed for Climate Adaptation? Landscape and Urban Planning 2012, 105, 27–33, doi:10.1016/j.landurbplan.2011.11.018.
  5. Adulkongkaew, T.; Satapanajaru, T.; Charoenhirunyingyos, S.; Singhirunnusorn, W. Effect of Land Cover Composition and Building Configuration on Land Surface Temperature in an Urban-Sprawl City, Case Study in Bangkok Metropolitan Area, Thailand. Heliyon 2020, 6, e04485, doi:10.1016/j.heliyon.2020.e04485.
  6. Liu, Y.; Peng, J.; Wang, Y. Diversification of Land Surface Temperature Change under Urban Landscape Renewal: A Case Study in the Main City of Shenzhen, China. Remote Sensing 2017, 9, 919, doi:10.3390/rs9090919.
  7. An, H.; Cai, H.; Xu, X.; Qiao, Z.; Han, D. Impacts of Urban Green Space on Land Surface Temperature from Urban Block Perspectives. Remote Sens. 2022, 14, 4580, doi:10.3390/rs14184580.
  8. Yang, C.; He, X.; Wang, R.; Yan, F.; Yu, L.; Bu, K.; Yang, J.; Chang, L.; Zhang, S. The Effect of Urban Green Spaces on the Urban Thermal Environment and Its Seasonal Variations. Forests 2017, 8, 153, doi:10.3390/f8050153.
  9. Chen, A.; Yao, X.A.; Sun, R.; Chen, L. Effect of Urban Green Patterns on Surface Urban Cool Islands and Its Seasonal Variations. Urban Forestry & Urban Greening 2014, 13, 646–654, doi:10.1016/j.ufug.2014.07.006.
  10. Wang, Y.; Chang, Q.; Li, X. Promoting Sustainable Carbon Sequestration of Plants in Urban Greenspace by Planting Design: A Case Study in Parks of Beijing. Urban Forestry & Urban Greening 2021, 64, 127291, doi:10.1016/j.ufug.2021.127291.
  11. Rakoto, P.Y.; Deilami, K.; Hurley, J.; Amati, M. Revisiting the Cooling Effects of Urban Greening: Planning Implications of Vegetation Types and Spatial Configuration. Urban For. Urban Green. 2021, 64, 127266, doi:10.1016/j.ufug.2021.127266.

Special thanks to you for your good comments.

Reviewer 3 Report

Summary of the manuscript:

This research aims to clarify the major factors affecting thermal effects on the micro-climates of urban parks and to propose practical adaptive park design approaches to mitigate urban heat island effects. It addresses the critical issue of urban heat island effect in rapidly developing, dense urban areas. The subject is especially relevant given rapid worldwide urbanization and climate change. The strength of the paper is the attempt to provide quantitative underpinnings that inform park design specifically and urban design more generally.

General comments:

While the introduction indicates the ultimate purpose of the research is to inform design, the focus of the article is on methodology and the specific methods. Fully twelve and one-half pages of this twenty page article are dedicated to methodology, method, and detailed quantitative results. This leads to reader fatigue and confusion. Is this research testing the efficacy of the methods, is it a description of the methods, or does it provide quantitative data to inform design? If it is the latter, as both the abstract and introduction claim, a suggestion would be to summarize the method in a few paragraphs followed by a summary of the quantitative results and move the exhaustive description of methods and quantitative results to an appendix. On the other hand, if the focus is on the method, that should be made clear so that the design implications take a subservient role: to test whether the method is sound. However, it is unclear whether that test was made.

The large number of abbreviates adds to reader fatigue and confusion. Can short descriptors be used instead?

The classifications of the landscape character and configuration are so finely differentiated as to be difficult to understand. Further, it is unclear whether the design recommendations are classified by landscape character and configuration or by landscape type. A clear and simpler classification (smaller number of classifications) would help. The definition gets lost in all the detail. Is this level of separation of landscape characters and configurations really relevant? Why? Does it correlate to design as practiced?

Finally, 4.3 Implications for urban landscape management, is also unclear. It is very difficult to understand how recommendations for one landscape type differ from another. It is unclear whether the data really justify this level of detailed differentiation by landscape type. Are you really simply saying add green and decrease impervious surfaces to all landscape types? If so, just say so and then describe how to do it.

The general impression is that the essential message of the research gets lost in the minutia of details. Attention to these details is necessary to conduct the research, but the purpose of an academic paper is to synthesize the information and reach generalizable conclusions that can be recognized and absorbed by a broader range of scholars – in this case the authors indicate the audience is designers. If that is the case, it needs to be more accessible.

Specific comments:

Study area, lines 135 through 147:

The abstract identifies 123 parks. The study area only identifies the city. Were there 123 parks? Are they all in the city of Luoyang? Does that include all parks in the city? If not, how were they chosen?

Figure 7 (this is true for all other charts and figures):

Do the charts reference specific parks? Most of the charts and diagrams are disconnected from important information the makes them relevant. They need to be organized by relevance.

Rating the Manuscript

  • Novelty: Is the question original and well-defined? Do the results provide an advancement of the current knowledge?
    • They could with significant revisions.
  • Scope: Does the work fit the journal scope*?
    • Yes.
  • Significance: Are the results interpreted appropriately? Are they significant? Are all conclusions justified and supported by the results? Are hypotheses carefully identified as such?
    • As currently presented, no. They could be relevant if the data are synthesized and generalized and the purpose of the manuscript is clarified.
  • Quality: Is the article written in an appropriate way? Are the data and analyses presented appropriately? Are the highest standards for presentation of the results used?
    • No, see general comments above.
  • Scientific Soundness: Is the study correctly designed and technically sound? Are the analyses performed with the highest technical standards? Is the data robust enough to draw conclusions? Are the methods, tools, software, and reagents described with sufficient details to allow another researcher to reproduce the results? Is the raw data available and correct (where applicable)?
    • The challenge is making the data relevant.
  • Interest to the Readers: Are the conclusions interesting for the readership of the journal? Will the paper attract a wide readership, or be of interest only to a limited number of people? (Please see the Aims and Scope of the journal.)
    • Not as written. The subject is valuable. With significant revisions it could be interesting.
  • Overall Merit: Is there an overall benefit to publishing this work? Does the work advance the current knowledge? Do the authors address an important long-standing question with smart experiments? Do the authors present a negative result of a valid scientific hypothesis?
    • It could with significant revisions.
  • English Level: Is the English language appropriate and understandable?
    • Yes.

Author Response

Response to the Review Comments

Dear reviewer,

Thank you for your insightful comments concerning our manuscript entitled “Discover the desirable landscape structure of urban parks for mitigating urban heat: a high spatial resolution study using a forest city, Luoyang, China as a lens”. Those comments are all valuable and very helpful for revising and improving our paper, as well as the important guiding significance to our research. We have studied comments carefully and have made corrections which we hope meet with approval. Revised portions are marked in red on the paper. The main corrections in the paper and the responses to your comments are as flowing:

[Comment #1]: While the introduction indicates the ultimate purpose of the research is to inform design, the focus of the article is on methodology and the specific methods. Fully twelve and one-half pages of this twenty page article are dedicated to methodology, method, and detailed quantitative results. This leads to reader fatigue and confusion. Is this research testing the efficacy of the methods, is it a description of the methods, or does it provide quantitative data to inform design? If it is the latter, as both the abstract and introduction claim, a suggestion would be to summarize the method in a few paragraphs followed by a summary of the quantitative results and move the exhaustive description of methods and quantitative results to an appendix. On the other hand, if the focus is on the method, that should be made clear so that the design implications take a subservient role: to test whether the method is sound. However, it is unclear whether that test was made.

[Response #1]: Thank you so much for your insightful comments. We are deeply sorry for the trouble caused to you, but please allow us to make a clear explanation.

1) This study provides an understanding of the major factors affecting the mitigation of thermal effects in urban parks. Using the land cover data, landscape pattern and LST in four seasons, which were quantified from WorldView-2 and Landsat-8 images, we investigated the relationship between LST and landscape features of different park categories, and obtained the theoretical results after integrated analysis of data, and based on the theory, inform the design of the parks. Basically, it's about providing quantitative data to inform design. So it is necessary to provide a clear description of the research methods to clear the way to obtain the results. In order to clarify the method and process, we have added a research flowchart (Figure 2) of the method and the flow in the method section.

2) This article is a research paper, and we thought removing the exhaustive description of quantitative results might cause the final part of the paper to appear unconvincing, due to a lack of data support. Moreover, a detailed description of the results can provide sufficient theoretical support for the guiding design part. So we discuss them in the implication for urban landscape management to mitigate urban heat and provide some pattern diagrams for urban park planning and design in section 4.3.

Thank you again for your valuable comments on our manuscript. We hope our correction and explanation could meet your approval.

[Comment #2]: The large number of abbreviates adds to reader fatigue and confusion. Can short descriptors be used instead?

[Response #2]: Thank you for pointing out the limitation of our work. Since the abbreviates contain a large number of landscape pattern indices, and the landscape pattern indices have agreed-upon abbreviations in previous papers. So we added "Abbreviations" after the abstract and keywords, and make a summary in the form of a table to facilitate readers understanding the article and reduce ambiguities.

[Comment #3]: The classifications of the landscape character and configuration are so finely differentiated as to be difficult to understand. Further, it is unclear whether the design recommendations are classified by landscape character and configuration or by landscape type. A clear and simpler classification (smaller number of classifications) would help. The definition gets lost in all the detail. Is this level of separation of landscape characters and configurations really relevant? Why? Does it correlate to design as practiced?

[Response #3]:

Thank you so much for your insightful comments. The design proposal is based on the existing park classification criteria for the parks in the study area. The purpose of classifying the parks according to the existing study criteria is to differentiate the results of the park classification through the classification in the renovation so that the renovation can be more directed.

The results of the paper are summarized within the categories and then applied to specific parks according to the park specifics. When designing a park in practice, the category of the park will be determined first based on the area, location, and surrounding environment. Therefore, this paper uses the categories of pre-construction classification of parks for discussion, so that the results can be directly correlated to the design in practice for the reference of designers.

Thank you again for your valuable comments on our manuscript. We hope our explanation could meet your approval.

[Comment #4]: Finally, 4.3 Implications for urban landscape management, is also unclear. It is very difficult to understand how recommendations for one landscape type differ from another. It is unclear whether the data really justify this level of detailed differentiation by landscape type. Are you really simply saying add green and decrease impervious surfaces to all landscape types? If so, just say so and then describe how to do it.

[Response #4]: Thank you for your valuable comments on our manuscript, and apologize for our unclear presentation.

The main purpose of this paper is to compare surface temperature differences in urban parks during different seasons and to analyze the driving factors. Based on the results derived from the study, to provide guidance for urban management in section 4.3. In our manuscript, we classified parks based on policies and used these classifications as the lowest-level basis for the description of the results. The purpose of the classification is, on the one hand, to clarify the differences in surface temperature and their drivers in various types of parks and, on the other hand, to better correspond to the categories of parks in order to make recommendations for urban planning. In section 4.3, at first, we recommend some suggestions for all parks such as increasing the proportion of vegetation then proposed some strategies for parks with higher LST such as PLP. In analyzing the drivers of LST in parks, some of the factors show high influence in different types of parks, therefore some strategies may be applied to multiple types of parks. This makes it appear that some of the strategies are relatively similar in the presentation of the article. This may be caused by the lack of clarity in our presentation of the manuscript, we have corrected them in section 4.3.

Once again, we apologize for the inconvenience caused by our inappropriate presentation. We hope our correction and explanation could meet your approval.

[Comment #5]: The general impression is that the essential message of the research gets lost in the minutia of details. Attention to these details is necessary to conduct the research, but the purpose of an academic paper is to synthesize the information and reach generalizable conclusions that can be recognized and absorbed by a broader range of scholars in this case the authors indicate the audience is designers. If that is the case, it needs to be more accessible.

[Response # Comment #5]:

Thank you so much for your insightful comments. What I want to explain here is that our manuscript is a research paper, and the main research content is to explore the seasonal variation of land surface temperature in urban parks and its driving factors. Material and methods describe the methods used in our study to quantify landscape patterns and land surface temperature and then explore the effects of landscape patterns on park LST through regression and correlation analysis. Then use the result to provide guidance for the park renovations.

In general, our recommendations are only suggestions for park construction, the main audience is scholars in related fields. Once again we are sorry for our inappropriate representation and the inconvenience for you and hopefully, our correction and explanation could meet your approval.

[Comment #6]: Study area, lines 135 through 147:

The abstract identifies 123 parks. The study area only identifies the city. Were there 123 parks? Are they all in the city of Luoyang? Does that include all parks in the city? If not, how were they chosen?

[Response # Comment #6]:

The 123 parks selected in our manuscript are all located in the urban area of Luoyang city (Figure 3). This data was obtained through a field investigation of parks in Luoyang in November 2020. We investigated all the parks that were completely built and in use in Luoyang at that time, and then remove the parks which are incomplete, with ambiguous positioning, and can not be found on the remote sensing image. Thank you for your valuable comments on our manuscript. We hope our correction and explanation could meet your approval.

Figure 3

[Comment #7]: Figure 7 (this is true for all other charts and figures):

Do the charts reference specific parks? Most of the charts and diagrams are disconnected from important information the makes them relevant. They need to be organized by relevance.

[Response # 7]:

The main research content and expected results of our manuscript are the seasonal land surface temperature differences in different parks and their influencing factors. Based on the results, some suggestions are made for municipal construction. Our manuscript mainly wants to derive results from the analysis and then make recommendations based on the results. It is mainly based on the results of the correlation analysis and stepwise regression analysis (Figure 8, 9) and corresponds to the recommendation in Table 3 to reduce the land surface temperature of urban parks and mitigate the urban heat effect. For example,area-mn in cp(Figure 8 a-d, 9 a-d) has a significant negative effect on the surface temperature in all seasons. In the proposal section, we believe that the surface temperature can be reduced by integrating the landscape to increase the area of the patches. We have modified the description of the recommendation in line 567-594. We hope our correction and explanation could meet your approval.

Special thanks to you for your good comments.

Round 2

Reviewer 3 Report

The added paragraphs provide better context for the research. The grammar in these paragraphs needs another edit.

I believe the earlier suggestion to move much of the methods section to an appendix would strengthen the paper, but will concede the point given the authors insistence on including them in the main text.

Author Response

Response to the Review Comments

Dear reviewer,

Thank you for your insightful comments concerning our manuscript entitled “Discover the desirable landscape structure of urban parks for mitigating urban heat: a high spatial resolution study using a forest city, Luoyang, China as a lens”. The comment is valuable and very helpful for revising and improving our paper, as well as the important guiding significance to our research. We have studied the comment carefully and have made corrections which we hope meet with approval. Revised portions are marked in red on the paper. The main corrections in the paper and the responses to your comments are as flowing:

[Comment #1]: The added paragraphs provide better context for the research. The grammar in these paragraphs needs another edit.

[Response #1]: Thank you so much for you carefully check. We are deeply sorry for the trouble caused to you. We have carefully revised the grammar of all parts of our manuscript and attached the revised version below. Thanks again for your suggestions.

Revised version:

The pace of urbanization worldwide is unprecedented, since reform and opening-up, China's urbanization rate has increased by 45.9%, much higher than the global average in the same period [1,2]. Rapid urbanization has increased heat storage in cities as the more vegetated area were replaced with underlying impervious surfaces[3] which leads to an exchange of energy between the urban surface and the atmosphere[4,5], so made that urban land surface temperature (LST) higher than surrounding non-urbanized areas, this phenomenon is defined as urban heat island (UHI) [6]. The UHI has been observed worldwide[7,8]. Industrial activities and anthropogenic heat sources have increased the scale and complexity of the UHI[9]. The intensification of UHI leads to abnormal changes in material and energy flows within cities and in the structure and function of ecosystems, so it seriously affects urban health and increases the risk of human violence and death [10–12]. Data from the China Meteorological Administration showed that the combined intensity of regional heat events in 2022 is the strongest since 1961, putting environmental pressure on Chinese cities[3]. Numerous empirical studies show that urban parks (UP) can cool and humidify the surrounding environment through evapotranspiration, shading, and water, it widely plays a leading role in reducing urban LST[13–15]. In the context of the background above, gaining a better understanding of the influencing factors of urban parks LST, enhancing the cooling effect, and maximizing the ecological benefits of UP is an important research direction and hot issue in improving the urban ecological environment and the city's ability to cope with climate change and essential to promote sustainable urban development[16–18].

LST is an essential parameter for studying the urban thermal environment and is also the main index to regulate the temperature of the urban bottom air and determine the surface radiation [6,19]. What is evident from the literature is that the widely used approaches for measuring urban climate and variations in LST can group into two major categories: Meteorological station monitoring data and Remote sensing data.

However, due to the limited geographic coverage of meteorological stations and the complexity of dynamic heat transfer processes, it frequently provides evidence of the impact of urbanization on climate warming at regional scales [20,21]. In contrast, with the rapid development of remote sensing and geospatial science, LST based on infrared remote sensing has been widely used in quantitative urban LST due to its relatively easy retrieval and collection, low cost, and reusability, therefore suitable for long-term continuity and related studies at different scales and accuracies [22].

The important influence of landscape structure on UHI has been widely demonstrated. An accurate assessment of land cover types is also essential for exploring the spatial pattern of UP and improving its quality [23]. Along with the advancement of remote sensing and geographic information system, land use/land cover (LULC) data has become a major source of monitoring spatial patterns in UP, especially studies at the city and regional levels [24]. To date, various raster-formatted LULC products have been widely applied in related research, and most of the quantitative land cover data used in the study have an accuracy of 30m and above. Therefore, many small UP may be ignored or misidentified as urban impervious pixels and excluded from studies under the constraint of satellite pixel accuracy[25], the landscape pattern of these parks is also hard to be accurately monitored due to the limitation of resolution. Small parks should not be neglected when quantifying the spatial extent and internal spatial pattern of parks, as they have been shown to have an effective cooling effect on the surrounding areas [26,27]. The distribution of landscape patterns within the parks needs to be estimated more accurately. In this paper, Word View-2 data with a resolution of 0.5m was used to monitor the LULC data of UP and cooperate with manual visual modification to improve the accuracy of the landscape pattern of UP.

Numerous studies have used remote sensing data to explore the change mechanism and scale of LST. In terms of urban climate, urban greening is often done to offset the negative impact of impervious surfaces on urban temperatures [28]. It has been found that UP can significantly reduce urban surface temperature [29,30]. The study of cities in different climate zones shows that the UP LST varies significantly across seasons and day and night[21,31,32]. As for driving factors, more attention has been paid to LULC, landscape structure, and energy consumption [33,34]. Many studies have confirmed that the ability of parks to regulate temperature is related to morphological indicators [35,36],. Gao found that the increased area and perimeter of parks can bring an enhanced cooling effect, and the shape index showed a relatively weak and inconsistent correlation with LST[37]. The effects of the park's spatial characteristics, such as the distribution, morphology, and composition of various landscapes, on the LST have also been confirmed in recent articles[38,39]. Maimaitiyiming et al discovered that increasing the PD and ED helps reduce LST[40]. Furthermore, the composition, quantity, type, horizontal and vertical structure, and proportion of the park's vegetation community all significantly impact the LST, trees have a stronger effect on LST than shrubs, with herbs playing the smallest role [41,42].

Nevertheless, due to the complex interaction of various factors and the spatial heterogeneity of cities, the causes of UHI are complex and site-specific, and the relationship between landscape structure and LST is inconsistent. A single measure cannot be effectively applied to all regions[12,43] and alleviating UHI by increasing the number of parks without limitation is unattainable due to the finite urban area[44]. Liu even thinks that inferences about factors influencing LST in global cities based on a limited number of studies may be invalid[45]. On this premise, accurately quantifying the relationship between landscape structure, LST, and its specific impact trend of the regional UP and seeking more sustainable, affordable measures is a meaningful and practical way to alleviate the risk of the urban thermal environment[46]. It is also critical for targeted improvements in urban landscape management and planning, as well as understanding the urban biophysical characteristics and processes required for long-term socioeconomic and environmental decision-making [45]. Studies on UHI in China are geographically biased, mainly focusing on metropolises with high urbanization rates and economically developed coastal cities, such as Shanghai, Beijing, Shenzhen, etc [47–50], with less focus on the less developed areas in the Central Plains and the northwest coast of China. In recent years, as one of the most crucial garden cities, forest cities, and economic and cultural centres in the Central Plains of China, Luoyang has experienced intense urbanization and rapid economic development, the city's borders have continued to expand, internal green space construction has continued to strengthen. At the same time, as the oldest city in the history of civilization and the eastern starting point of the Silk Road, Luoyang has a profound background in garden construction and occupies an undoubted cultural position, it is one of the most representative cities in the Central Plains. Therefore, it is imperative to explore the relationship between the landscape pattern and the LST of UP in Luoyang and find the optimal planning mode of UP for the economic and cultural construction and ecological improvement of Luoyang.

Therefore, this study conducted an accurate definition of the landscape pattern base on the high-resolution remote sensing data and a comprehensive comparison of the seasonal variation of the effects of landscape pattern on park LST by specific park categories to puts forward the concrete and implementable urban planning scheme for urban management with the focus on (1) Use high-resolution remote sensing data to obtain UP landscape types and their shape indicators accurately. (2) Retrieve LST from remote sensing satellite infrared data, and probe the relationship between LST and park morphological indicators and landscape patterns. (3) Analyze the maximization mode of LST regulation of Luoyang UP, and provide targeted recommendations and models for urban planning on mitigate the UHI effects through the rational allocation of the interior landscape of the park.

Sincerely thank you for your good comments.
